# YTHDF3 recognizes DNA N6-methyladenine and recruits ALKBH1 for 6mA removal from genomic DNA

Xin-Hui Chen [1,3], Zi-Lu Wang[1,3], Jincui Yang[1,3], Min Chen [1,3], Si-Yi Zhao[1], Kun-Xiong Guo[1], Xuelong Zheng[1], Zhengwei Zhao[1], Xiaoqiang Chen[1], Jing Li [1], Min-Min Zhang[1], Ling Ran[1], Huifang Zhu[1], Xiao-Feng Gu [2✉] & Guang-Rong Yan [1✉]

## Abstract

DNA N⁶-methyladenine (6mA) is an emerging epigenetic mark in the mammalian genome. ALKBH1 preferentially exhibits 6mA demethylase activity for single-stranded DNA (ssDNA) or bubbled/bulged DNA, but not for double-stranded DNA (dsDNA). Nevertheless, ALKBH1 significantly decreases the cellular 6mA level in genomic DNA, whose prevailing DNA conformation in living mammalian cells is dsDNA. Therefore, the demethylase activity of ALKBH1 toward 6mA in genomic DNA, especially dsDNA, remains largely debated. Here, we found that YTHDF3 increases the 6mA demethylase activity of ALKBH1 in genomic DNA with different conformations, including dsDNA. Compared with ALKBH1, YTHDF3 preferentially recognizes and binds to 6mA-modified DNA with different conformations. YTHDF3 recognizes 6mA in genomic DNA, and binds ALKBH1 to recruit it to sites near 6mA in genomic DNA, thereby facilitating the ALKBH1-mediated removal of 6mA in genomic dsDNA. In summary, YTHDF3 is a novel genomic DNA reader and guides ALKBH1 to remove 6mA in human genomic DNA.

**Keywords** Genomic DNA 6mA; ALKBH1; Genomic 6mA Reader; YTHDF3; dsDNA
**Subject Categories** Chromatin, Transcription & Genomics; DNA Replication, Recombination & Repair

## Introduction

DNA N⁶-methyladenine (6mA) modification was originally found in the genome of prokaryotes and is the most abundant modification in most bacterial genomic DNA related to restriction-modification systems, DNA repair, DNA replication and transcription regulation. Recent studies have confirmed the presence of 6mA modifications in the genomes of several eukaryotic species, including mammals such as humans (Xiao et al, 2018; Xie et al, 2018). However, the presence and abundance of 6mA in the mammalian genome remains a subject of considerable debate because 6mA modification is relatively rare in the mammalian genome (Douvlataniotis et al, 2020; Kong et al, 2022). With the development of methodologies for highly sensitive detection and sequencing of genome 6mA (Chen et al, 2022; Feng et al, 2024; Koh et al, 2018; Li et al, 2022), an increasing number of studies have supported the presence of genome 6mA modifications in eukaryotes, including mammals, but the abundance of genomic 6mA varies in different cells and tissues and is dynamically altered during embryonic development and processes related to disease progression, such as tumorigenesis (Boulias and Greer, 2022; Shen et al, 2022; Xiao et al, 2018). Although the frequency of 6mA is low in the mammalian genome, it is a novel epigenetic mark and has been implicated in embryonic development, cancer, the mitochondrial stress response and adaptation, and stem cell differentiation (Feng and He, 2023; Hao et al, 2020; Li et al, 2020b; Lyu et al, 2022; Ma et al, 2018; Shen et al, 2022; Wu et al, 2016; Xiao et al, 2018).

The deposition and erasure of epigenetic modifications are catalyzed primarily by enzymes. Substantial efforts have been made to search for mammalian genome DNA 6mA methyltransferases (writers) and demethylases (erasers) in eukaryotes, although besides methylase-deposited 6mA, several studies have reported the presence of misincorporated 6mA in cells (Chen et al, 2023; Liang et al, 2024; Liu et al, 2020). However, such explorations are challenging due to the low abundance of 6mA in genomic DNA and the lack of good and functional research models. Currently, 6mA methyltransferases and demethylases have been identified in only a few eukaryotes. DAMT-1 in *C. elegans* (Greer et al, 2015), METTL4 in *M. musculus* and *H. sapiens* (Hao et al, 2020; Kweon et al, 2019), and N6AMT1 in *H. sapiens* (Xiao et al, 2018) have been shown to be DNA 6mA methyltransferases. However, although N6AMT1 has been reported to regulate genomic 6mA in multiple mammalian cells (Li et al, 2019b; Xiao et al, 2018), its 6mA methyltransferase activity in vitro has been disputed (Li et al, 2019a; Metzger et al, 2019; Woodcock et al, 2019);

[1]Biomedicine Research Center, Guangdong Provincial Key Laboratory of Major Obstetric Disease, Guangdong Provincial Clinical Research Center for Obstetrics and Gynecology, The Third Affiliated Hospital, Guangzhou Medical University, 510150 Guangzhou, China. [2]Biotechnology Research Institute, Chinese Academy of Agricultural Sciences, 100081 Beijing, China. [3]These authors contributed equally: Xin-Hui Chen, Zi-Lu Wang, Jincui Yang, Min Chen. ✉E-mail: guxiaofeng@caas.cn; tgryan@jnu.edu.cn

And the role of METTL4 as a DNA methyltransferase is also controversial (Bochtler and Fernandes, 2020). NMAD-1 in *C. elegans* (Zhang et al, 2015), Dmad in *D. melanogaster* (Zhang et al, 2015), ALKBH1 in *M. musculus* and *H. sapiens* (Wu et al, 2016; Xiao et al, 2018), and ALKBH4 in *M. musculus* (Kweon et al, 2019) have been shown to be DNA 6mA demethylases. However, although ALKBH4 displays 6mA demethylation activity on dsDNA in vitro, the activity is very weak and requires further in vitro and cell-based support; And the role of DMAD/TET as a 6mA demethylase is also controversial (Boulet et al, 2023). Therefore, currently, genomic DNA 6mA methyltransferases and demethylases are still unidentified in most eukaryotes. In addition, the mechanisms by which DNA 6mA methyltransferases and demethylases regulate 6mA methylation and demethylation remain largely unexplored.

Our study and other reports reveal that ALKBH1 is a DNA 6mA demethylase in mammals (Wu et al, 2016; Xiao et al, 2018; Xie et al, 2018). Subsequent structural analyses indicated that ALKBH1 has 6mA demethylase activity with a preference for single-stranded DNA (ssDNA) or bubbled/bulged DNA but not for double-stranded DNA (dsDNA) (Tian et al, 2020; Zhang et al, 2020). However, ALKBH1 can significantly decrease the 6mA level in cellular genomic DNA (Wu et al, 2016; Xiao et al, 2018; Xie et al, 2018), whose prevailing conformation in living mammalian cells is dsDNA. Therefore, the demethylase activity of ALKBH1 toward 6mA in genomic DNA, especially dsDNA, remains debated.

The biological functions and importance of 6mA modification in genomic DNA are dependent on 6mA-binding proteins (that is, 6mA readers), which control genomic DNA fate and function to influence distinct biological functions. However, to date, in eukaryotes, only Jumu in *D. melanogaster*, SSBP1 in *H. sapiens* and YTHDC1 bind to 6mA in DNA or mitochondrial DNA (mtDNA) (He et al, 2019; Koh et al, 2018; Woodcock et al, 2020; Yu et al, 2021). Moreover, the 6mA modification in DNA disrupts the binding of SATB1 in *M. musculus* and TFAM in *H. sapiens* mitochondria to DNA and mtDNA, respectively (Hao et al, 2020; Li et al, 2020b). Genomics DNA 6mA readers in most eukaryotes, including humans, remain largely unknown.

In this study, we found that YTHDF3 increases the demethylase activity of ALKBH1 toward 6mA in genomic DNA, including dsDNA. YTHDF3 is a novel genomic DNA 6mA reader in mammals. YTHDF3 recognizes 6mA in genomic DNA, including dsDNA, and binds to ALKBH1 to recruit ALKBH1 to sites near 6mA modifications in genomic DNA to facilitate the ALKBH1-mediated removal of 6mA in genomic DNA including dsDNA.

## Results

### YTHDF3 decreases the level of 6mA in cellular genomic DNA

When we investigated the 6mA demethylase activity of ALKBH1 on genomic DNA by an in vitro demethylation assay, interestingly and surprisingly, we found that ALKBH1 immunopurified from human cells efficiently decreased the level of 6mA in synthetic 6mA-modified dsDNA, ssDNA, bulged DNA and bubbled DNA oligonucleotides, whereas as in previous reports (Tian et al, 2020; Zhang et al, 2020), recombinant ALKBH1 purified from bacteria decreased the level of 6mA in synthetic 6mA-modified ssDNA,

bulged DNA and bubbled DNA but not in dsDNA (Fig. 1A–C; Appendix Fig. S1A). This finding suggests that proteins in the immunopurified ALKBH1 complex may regulate the demethylase activity of ALKBH1 toward 6mA in genomic DNA, especially in dsDNA.

Furthermore, the immunopurified ALKBH1 complex was dissociated, and the proteins that interact with ALKBH1 in this complex were identified via mass spectrometry (Fig. 1D). A total of 335 proteins that interact with ALKBH1 were identified (Dataset EV1). Given that ALKBH1 is a genomic 6mA demethylase whose activity is dependent on $Fe^{2+}$ and 2-oxoglutarate, the proteins associated with RNA m6A, methyl and iron ions were first selected for further investigation. Therefore, YTHDF3, MTHFD1 and MMS-19 were preliminarily selected for further investigation.

We applied 6mA-RE-qPCR to determine the change in the level of cellular genomic DNA 6mA induced by ALKBH1 and/or other factors as previously described (Fu et al, 2015; Xiao et al, 2018), because the 6mA-RE-qPCR assay can avoid the problems associated with the low abundance of 6mA in cellular genomic DNA, potential prokaryotic DNA contamination in samples, and potential contamination of RNA m6A in DNA. We selected 4 methylated 6mA sites with GATC motifs (M2, M3, M4, and M5 in Supplementary S1E in our previous reports (Xiao et al, 2018)) as representative for determining the changes in the level of 6mA in cellular genomic DNA caused by ALKBH1 or other factors after the feasibility of the 6mA-RE-qPCR assay for determining changes in the DNA 6mA level was validated using two standards (Appendix Fig. S1B). We found that silencing YTHDF3 increased the level of cellular genomic DNA 6mA in human HTC-116 cells, whereas silencing MTHFD1 and MMS-19 did not change the level of cellular genomic 6mA (Appendix Fig. S1C,D). Similar results were observed in YTHDF3-silenced HEK293T and HeLa cells and in YTHDF3 knockout (KO) HeLa cells (Figs. 1E and 2A,B; Appendix Fig. S1E). The overexpression of YTHDF3 decreased the level of cellular genomic DNA 6mA in HEK-293T and HeLa cells (Fig. 1F; Appendix Fig. S1F). Therefore, YTHDF3 was deemed of interest and was the focus of this study.

Finally, neither YTHDF3 immunopurified from human ALKBH1 KO cells nor recombinant YTHDF3 purified from bacteria directly demethylated 6mA in synthetic 6mA-modified dsDNA (6mA-dsDNA) oligonucleotide substrates in the in vitro demethylation assay (Fig. 1G,H), whereas YTHDF3 immunopurified from human ALKBH1-expressing cells decreased the 6mA level in 6mA-dsDNA oligonucleotide substrates in a dose-dependent manner (Fig. 1G), indicating that YTHDF3 does not have intrinsic 6mA demethylase activity but possibly participates in the regulation of the demethylase activity of ALKBH1 toward 6mA in genomic DNA, including dsDNA.

### YTHDF3 increases the demethylase activity of ALKBH1 toward 6mA in genomic DNA, including dsDNA

The overexpression of ALKBH1 decreased the cellular DNA 6mA level in YTHDF3-expressing cells, whereas the reduction in the cellular DNA 6mA level induced by the overexpression of ALKBH1 was blocked when YTHDF3 expression was knocked out in cells (Fig. 2A; Appendix Fig. S2A). The overexpression of YTHDF3 reduced the cellular DNA 6mA level in the ALKBH1-expressing HeLa cells, whereas the overexpression of YTHDF3 did not reduce

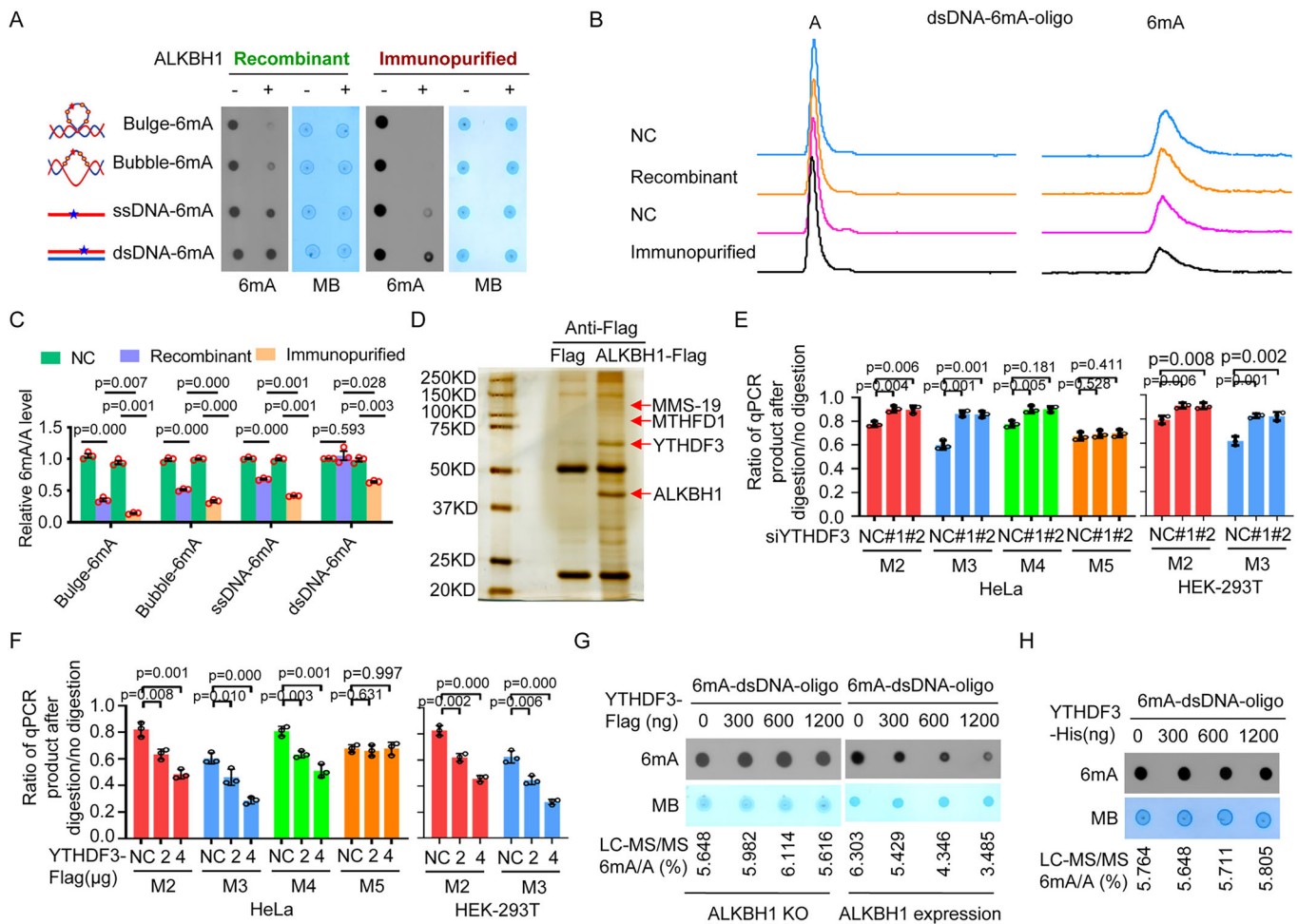

**Figure 1. YTHDF3 decreases the level of 6mA in cellular genomic DNA.**

(A, C) Recombinant ALKBH1 purified from bacteria or ALKBH1 immunopurified from human cells was incubated with 6mA-modified DNA (6mA-DNA) oligos with different conformations in the in vitro demethylase reaction. DNA 6mA levels were determined by dot blotting (A) and LC–MS/MS (C). (B) Representative LC–MS/MS spectrum for analyzing 6mA in the 6mA-dsDNA oligos mediated by recombinant ALKBH1 and immunopurified ALKBH1. (D) ALKBH1-FLAG complexes were coimmunoprecipitated with an anti-FLAG antibody, separated via SDS–PAGE and stained with silver; then, the proteins in the differential gel bands were identified via mass spectrometry. (E) Two independent siRNAs targeting YTHDF3 were transfected into HEK293T and HeLa cells, and cellular genomic DNA 6mA levels were determined by RE-6mA-qPCR. (F) The indicated concentrations of the YTHDF3 plasmids were transfected into HEK293T and HeLa cells, and the cellular genomic DNA 6mA levels were determined via RE-6mA-qPCR. (G) YTHDF3 immunopurified from ALKBH1-expressing or ALKBH1-KO HeLa cells was incubated with 6mA-dsDNA oligos, and the 6mA levels were determined by dot blotting and LC–MS/MS. (H) Recombinant YTHDF3 purified from bacteria was incubated with 6mA-dsDNA oligos, and the 6mA levels were determined. Data information: In (C, E, F), data are presented as mean ± SD. $n = 3$ independent biological replicates, *$P < 0.05$, **$P < 0.01$, ***$P < 0.001$, ns non-significant (Student's $t$ test). Source data are available online for this figure.

the cellular DNA 6mA level in the ALKBH1-KO HeLa cells (Appendix Fig. S2B,C). Cooverexpression of YTHDF3 increased the ALKBH1 overexpression-induced reduction in the cellular genomic 6mA level (Fig. 2B; Appendix Fig. S2D).

In the in vitro DNA 6mA demethylation assay, ALKBH1 immunopurified from YTHDF3-expressing cells directly and efficiently decreased the 6mA level in 6mA-dsDNA oligonucleotide substrates, whereas ALKBH1 immunopurified from YTHDF3 KO cells did not (Fig. 2C). ALKBH1 immunopurified from YTHDF3 KO cells did not remove 6mA from the 6mA-dsDNA oligonucleotide substrates, whereas YTHDF3 directly and efficiently increased the eraser activity of ALKBH1 toward 6mA from 6mA-dsDNA oligonucleotide substrates (Fig. 2D). Furthermore, as expected, recombinant ALKBH1 did not remove 6mA from the 6mA-dsDNA

oligonucleotide substrates, whereas recombinant YTHDF3 directly and efficiently promoted the eraser activity of recombinant ALKBH1 toward 6mA from the 6mA-dsDNA oligonucleotide substrates during in vitro demethylation (Fig. 2E; Appendix Fig. S2E). Our previous findings revealed that the DNA 6mA demethylase activity of ALKBH1 was dependent on $Fe^{2+}$ and 2-oxoglutarate (Xiao et al, 2018). Here, we found that the stimulatory effects of YTHDF3 on the dsDNA 6mA demethylase activity of ALKBH1 were dependent on $Fe^{2+}$ and 2-oxoglutarate (Appendix Fig. S2F).

The effects of YTHDF3 on the demethylase activity of ALKBH1 toward 6mA in ssDNA were also investigated. Both ALKBH1 immunopurified from YTHDF3 KO cells and recombinant ALKBH1 removed 6mA from 6mA-ssDNA oligonucleotide

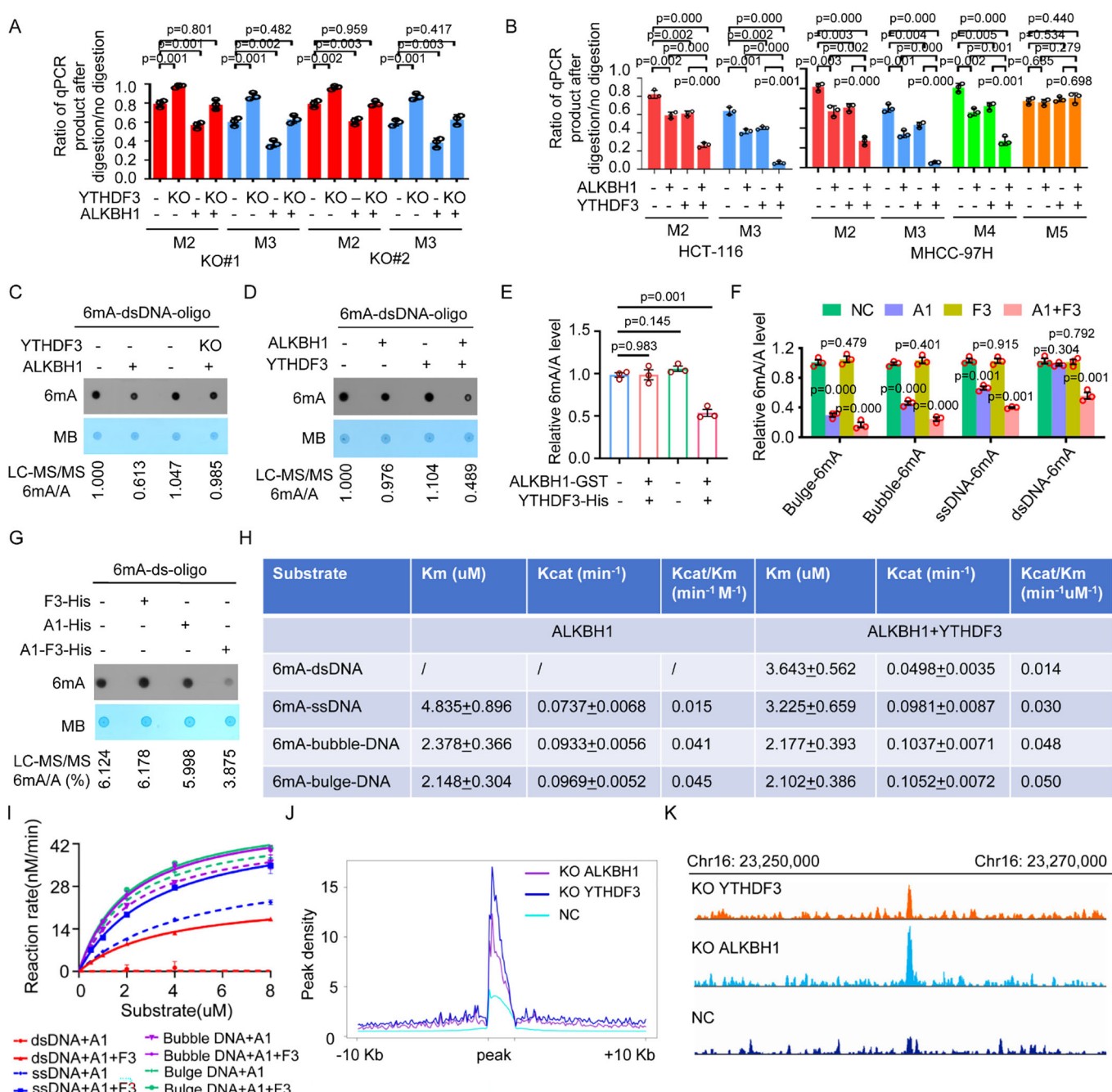

**Figure 2. YTHDF3 directly increases the 6mA demethylase activity of ALKBH1 toward 6mA in genomic DNA, including dsDNA.**

(**A**) ALKBH1 plasmids were transfected into YTHDF3-expressing or YTHDF3-KO HeLa cells, and cellular genomic DNA 6mA levels were determined by RE-6mA-qPCR. (**B**) ALKBH1 plasmids and YTHDF3 plasmids were cotransfected into HCT-116 and MHCC-97H cells, and cellular genomic DNA 6mA levels were determined by RE-6mA-qPCR. (**C**) ALKBH1 immunopurified from YTHDF3-expressing or YTHDF3 KO HeLa cells was incubated with 6mA-dsDNA oligos, and the 6mA levels were determined. (**D**) ALKBH1 immunopurified from YTHDF3 KO HeLa cells and YTHDF3 immunopurified from ALKBH1 KO HeLa cells were incubated with 6mA-dsDNA oligos, and the 6mA levels were determined. (**E**) Recombinant ALKBH1 and recombinant YTHDF3 were incubated with 6mA-dsDNA oligos, and the 6mA levels were determined by LC–MS/MS. (**F**) Recombinant ALKBH1 (A1) and recombinant YTHDF3 (F3) were incubated with 6mA-modified DNA oligos with different conformations, and the 6mA levels were determined by LC–MS/MS. (**G**) Recombinant ALKBH1 (A1), YTHDF3 (F3) and the ALKBH1/YTHDF3 (A1-F3) complex were incubated with 6mA-modified dsDNA oligos, and the 6mA levels were determined. (**H, I**) Steady-state kinetic constants (**H**) and Michaelis–Menten plot of the steady-state kinetics (**I**) for demethylation of 6mA in DNA with different conformations by ALKBH1 and ALKBH1 together with YTHDF3. (**J**) Cumulative distribution curve for the 6mA peak abundance on genomic DNA in NC, ALKBH1 KO (KO A1) and YTHDF3 KO (KO F3) HeLa cells, as determined by 6mA-IP-seq. (**K**) Representative overlapping regions of 6mA-modified DNA peaks in NC, ALKBH1 KO and YTHDF3 KO HeLa cells. Data information: In (**A, B, E, F, H**), data are presented as mean ± SD. $n = 3$ independent biological replicates, *$P < 0.05$, **$P < 0.01$, ***$P < 0.001$, ns non-significant (Student's $t$ test). Source data are available online for this figure.

substrates (Appendix Fig. S3). In addition, both YTHDF3 immunopurified from ALKBH1 KO cells and recombinant YTHDF3 directly and efficiently increased the eraser activity of immunopurified ALKBH1 and recombinant ALKBH1 toward 6mA from 6mA-ssDNA oligonucleotide substrates (Appendix Fig. S3). Furthermore, the effects of YTHDF3 on the demethylase activity of ALKBH1 toward 6mA in DNA with different conformations were investigated. Recombinant YTHDF3 directly increased the eraser activity of recombinant ALKBH1 toward 6mA in various synthetic DNA substrates with different conformations, including dsDNA, ssDNA, and bulged or bubbled DNA oligonucleotides (Fig. 2F; Appendix Fig. S2G). Finally, ALKBH1 and YTHDF3-His were coexpressed in bacteria, and the recombinant ALKBH1/YTHDF3-His complex (A1-F3-His) was copurified (Appendix Fig. S1G,H). We found that neither recombinant ALKBH1 alone nor recombinant YTHDF3 alone removed 6mA from the dsDNA substrate, while the recombinant ALKBH1/YTHDF3 complex markedly removed 6mA from the dsDNA substrate (Fig. 2G).

The steady-state kinetics analyses showed that ALKBH1 alone did not remove the 6mA modification in dsDNA, while ALKBH1 together with YTHDF3 removed the 6mA modification in 6mA-dsDNA with $K_{cat} = 0.0498$ min$^{-1}$ and $K_m = 3.643$ μM (Fig. 2H,I). And YTHDF3 also increased the 6mA demethylation activity of ALKBH1 toward 6mA-ssDNA with $K_{cat}/K_m$ from 0.015 min$^{-1}$ μM$^{-1}$ to 0.030 min$^{-1}$ μM$^{-1}$, 6mA-bubble DNA with $K_{cat}/K_m$ from 0.041 min$^{-1}$ μM$^{-1}$ to 0.048 min$^{-1}$ μM$^{-1}$ and 6mA-bluge DNA with $K_{cat}/K_m$ from 0.045 min$^{-1}$ μM$^{-1}$ to 0.050 min$^{-1}$ μM$^{-1}$ (Fig. 2H,I).

We next performed 6mA-IP-seq in ALKBH1 KO and YTHDF3 KO cells to compare the 6mA-enriched genomic regions regulated by ALKBH1 and YTHDF3. The number of 6mA peaks increased from 7403 in negative control (NC) cells to 11177 in ALKBH1 KO cells and to 15834 in YTHDF3 KO cells, indicating that 6mA modification at more sites and greater 6mA abundance in genomic DNA resulted from ALKBH1 or YTHDF3 KO (Fig. 2J,K). The 6mA peaks in ALKBH1 KO cells highly overlapped with those in YTHDF3 KO cells (Fig. 2K), suggesting that ALKBH1 and YTHDF3 regulate the same 6mA sites in genomic DNA. Consistent with our previous findings in the human leukomonocyte genome (Xiao et al, 2018), 6mA peaks were enriched in the exonic regions of the genome (Appendix Fig. S4A) and in the mitochondrial genome (Appendix Fig. S4B,C). As noted in previous reports (Xiao et al, 2018), most of the 6mA-modified DNA peaks were located in intronic and intergenic regions of the genome (Appendix Fig. S4D), and the [G/C]AGG[C/T] motif was the most prevalent motif in the 6mA sites in the human genome (Appendix Fig. S4E). Taken together, these findings indicate that YTHDF3 directly increases the demethylase activity of ALKBH1 toward 6mA in genomic DNA, including dsDNA.

## YTHDF3 increases the binding of ALKBH1 to 6mA in genomic DNA, including dsDNA

To investigate how YTHDF3 affects the genomic DNA 6mA demethylase activity of ALKBH1, the influence of YTHDF3 on the binding of ALKBH1 to 6mA in dsDNA was investigated. YTHDF3 increased the binding of cellular ALKBH1 to 6mA in dsDNA in a dose-dependent manner (Fig. 3A). The binding of ALKBH1 to 6mA in dsDNA was decreased when YTHDF3 expression was knocked out in cells but was restored when YTHDF3 was re-expressed in YTHDF3-silenced cells (Fig. 3B). The binding of ALKBH1 to 6mA in dsDNA was increased when ALKBH1 was overexpressed in YTHDF3-expressing cells but was not altered when ALKBH1 was overexpressed in YTHDF3 KO cells (Fig. 3C). A similar result was obtained in YTHFD3-silenced cells (Fig. 3D).

Furthermore, YTHDF3 directly and efficiently increased the binding of ALKBH1 to 6mA in dsDNA (Fig. 3E). Moreover, YTHDF3 directly increased the binding of ALKBH1 to 6mA in bulged DNA, bubbled DNA, dsDNA and ssDNA (Fig. 3F). Collectively, these findings indicate that YTHDF3 directly increases the binding of ALKBH1 to 6mA in genomic DNA, including dsDNA.

## YTHDF3, not YTHDF1/2 or YTHDC1/2, directly and specifically binds to ALKBH1

As shown by our interactome data in Fig. 1D and Dataset EV1, the interaction of YTHDF3 with ALKBH1 was validated (Fig. 4A). Given that YTHDF3 and ALKBH1 can bind to 6mA-modified DNA, to investigate whether the interaction of YTHDF3 with ALKBH1 is dependent on DNA, ALKBH1-YTHDF3 complexes were treated with DNase. We further confirmed the interaction of YTHDF3 with ALKBH1 in the presence of DNase, indicating that this interaction was DNA independent (Fig. 4B). Furthermore, the results of the GST pull-down assay revealed that YTHDF3 directly bound to ALKBH1 (Fig. 4C).

ALKBH1 consists of an N-terminal extension (NTE), a nucleotide recognition lid (NRL), and a C-terminal double-stranded β helix (DSBH) domain (Zhang et al, 2020). We found that constructs containing only the DSBH domain but not those containing only the other regions of ALKBH1 retained the ability to interact with YTHDF3 (Appendix Fig. S5). YTHDF3 consists of a YTH domain and a P/Q/N-rich domain (Fig. 4D). We found that the YTH domain of YTHDF3 is essential for ALKBH1 binding (Fig. 4E). All the YTHDF1/2/3 and YTHDC1/2 proteins contain a YTH domain (Murakami and Jaffrey, 2022). Do all the YTHDF1/2/3 and YTHDC1/2 proteins also bind to ALKBH1? Surprisingly, we found that only YTHDF3 binds to ALKBH1, whereas YTHDF1, YTHDF2 and YTHDC1 do not (Fig. 4F), which is consistent with our interactome data, in which YTHDF1/2 and YTHDC1/2 were not identified to interact with ALKBH1 (Fig. 1D; Dataset EV1), suggesting that a dissimilarity in the YTH domain of YTHDF3 may control the preferential binding of YTHDF3 to ALKBH1 relative to YTHDF1/2 and YTHDC1/2.

We further found that seven residues in the YTH domain of YTHDF3 are different from those in the YTH domain of YTHDF1 and YTHDF2 and that the amino acid sequence of the YTH domain of YTHDF3 is markedly different from those of the YTH domains of YTHDC1 and YTHDC2 (Fig. 4G). Mutation of these seven residues (L455M/L461V/V482A/A487T/Y488C/E502D/K504R, hereafter denoted MTbind) completely abolished the interaction of YTHDF3 with ALKBH1 (Fig. 4H). However, the recombinant YTHDF3 MTbind mutant retained the ability to bind to 6mA in dsDNA (Fig. 4I,J). A similar result was obtained for the immunopurified YTHDF3 MTbind mutant (Fig. 4K). Therefore, the YTHDF3 MTbind mutant binds to 6mA in dsDNA but does not bind to ALKBH1. Taken together, these results indicate that YTHDF3 directly and specifically binds to ALKBH1, whereas the

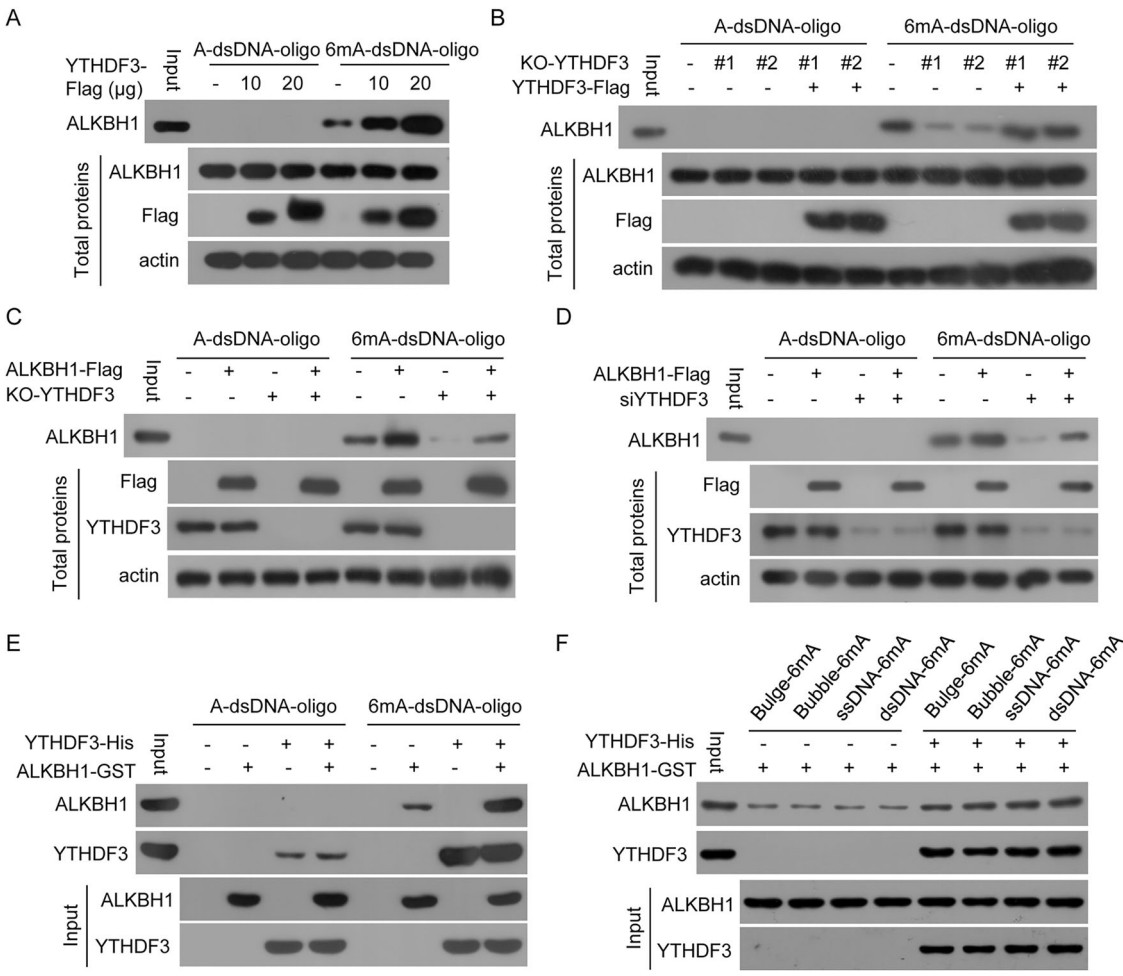

**Figure 3. YTHDF3 directly increases the binding of ALKBH1 to 6mA in genomic DNA, including dsDNA.**

(A) The indicated amounts of the YTHDF3 plasmid were transfected into HeLa cells, and the binding of cellular ALKBH1 to 6mA-modified and unmodified dsDNA oligos was detected via a DNA pull-down assay. (B) YTHDF3 plasmids were transfected into YTHDF3 KO HeLa cells, and the binding of cellular ALKBH1 to 6mA-modified and unmodified dsDNA oligos was detected. (C) ALKBH1 plasmids were transfected into YTHDF3-expressing or YTHDF3 KO HeLa cells, and the binding of cellular ALKBH1 to 6mA-modified and unmodified dsDNA oligos was detected. (D) ALKBH1 plasmids and siRNAs targeting YTHDF3 were cotransfected into HeLa cells, and the binding of cellular ALKBH1 to 6mA-modified and unmodified dsDNA oligos was detected. (E) Recombinant YTHDF3 and/or recombinant ALKBH1 were incubated with 6mA-modified and unmodified dsDNA oligos, and the binding of ALKBH1 and YTHDF3 to the 6mA-modified and unmodified dsDNA oligos was detected. (F) Recombinant YTHDF3 and/or recombinant ALKBH1 were incubated with 6mA-methylated DNA oligos with different conformations, and the binding of ALKBH1 and YTHDF3 to 6mA-DNA was detected. Source data are available online for this figure.

other YTH family proteins (YTHDF1, YTHDF2 and YTHDC1/2) do not bind to ALKBH1.

## YTHDF3 recognizes and binds to 6mA in human genomic DNA, including dsDNA

Previous studies have shown that YTHDF3, as an RNA m$^6$A reader, recognizes and binds to m$^6$A on RNA through its YTH domain(Chang et al, 2020; Fu and Zhuang, 2020; Li et al, 2020a; Shi et al, 2017). We found that recombinant YTHDF3 had a stronger ability to bind to 6mA-modified dsDNA than to 6mA-unmodified dsDNA (Fig. 5A). Similar results were observed for YTHDF3 immunopurified from ALKBH1 KO cells (Appendix Fig. S6A). YTHDF3 had stronger binding to 6mA-modified DNA with various conformations, including bubbled DNA, bulged DNA,

ssDNA and dsDNA, than did 6mA-unmodified DNA (Fig. 5B,C). YTHDF3 preferentially bound to 6mA in dsDNA and m$^6$A in RNA but did not bind to m$^1$A in tRNA (Fig. 5D). Furthermore, we used surface plasmon resonance (SPR) to analyze the binding ability of ALKBH1 and YTHDF3 to 6mA-modified and -unmodified dsDNA and found that the KD values of YTHDF3 to 6mA-modified dsDNA and 6mA-unmodified dsDNA were 161.7 nM and 512.1 nM, respectively, and the KD values of ALKBH1 to 6mA-modified dsDNA and 6mA-unmodified dsDNA were 1239 nM and 2441 nM, respectively (Fig. 5E). These results indicated that the binding ability of YTHDF3 and ALKBH1 to 6mA-modified dsDNA is greater than that to 6mA-unmodified dsDNA and that the binding ability of YTHDF3 to 6mA-modified dsDNA is greater than that of ALKBH1. Taken together, YTHDF3 preferentially recognizes and binds to 6mA in DNA.

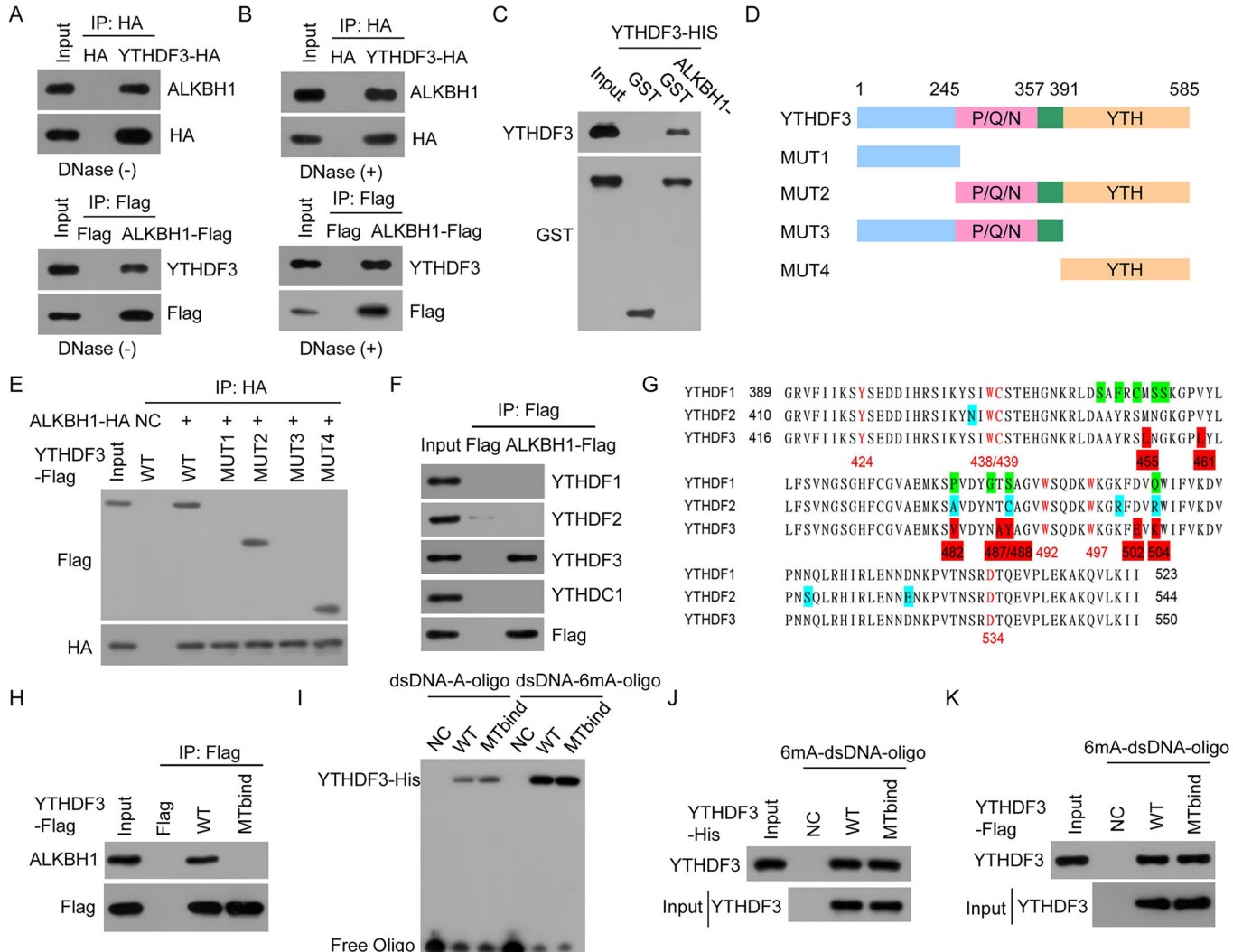

**Figure 4. ALKBH1 directly and specifically binds to YTHDF3, not YTHDF1/2 or YTHDC1/2.**

(A, B) YTHDF3-HA and ALKBH1-Flag plasmids were transfected into HeLa cells. YTHDF3-HA and ALKBH1-Flag complexes were coimmunoprecipitated via anti-HA and anti-FLAG antibodies in the absence (A) and presence (B) of DNase treatment, and endogenous ALKBH1 and YTHDF3 in these complexes were detected. (C) The direct binding of recombinant ALKBH1 to recombinant YTHDF3 was analyzed via a GST pull-down assay. (D) Diagram of wild-type YTHDF3 and its mutants with different domains. (E) The interactions of ALKBH1 with the indicated YTHDF3 mutants were detected. (F) The interactions of ALKBH1 with YTHDF1, YTHDF2, YTHDF3 and YTHDC1 were detected. (G) Comparison of the amino acid sequences of the YTH domain in YTHDF1, YTHDF2 and YTHDF3. (H) The interactions of wild-type YTHDF3 and the YTHDF3 MTbind mutant with endogenous ALKBH1 were detected. (I) The binding of recombinant wild-type YTHDF3 and the YTHDF3 MTbind mutant to 6mA-modified and -unmodified dsDNA oligos was determined by EMSA. (J) The binding of recombinant wild-type YTHDF3 and the YTHDF3 MT-binding mutant to 6mA-dsDNA oligos was determined by DNA pulldown. (K) The binding of wild-type YTHDF3 and the YTHDF3 MTbind mutant immunopurified from ALKBH1 KO HeLa cells to 6mA-dsDNA oligos was evaluated. Source data are available online for this figure.

To investigate which domain of YTHDF3 recognizes and binds to 6mA in genomic DNA, the binding of mutants containing different domains of YTHDF3 to the 6mA-dsDNA was investigated. We found that the YTH domain of YTHDF3 recognizes and binds to 6mA in genomic DNA (Appendix Fig. S6B). Previous studies have indicated that two residues in the YTH domain, W438 and W492, within the hydrophobic pocket of YTHDF3 contribute to the specific recognition of m⁶A in RNA (Chang et al, 2020). We found that, relative to that of wild-type YTHDF3, the binding ability of the YTHDF3 W438R/W492R mutant (hereafter denoted MT6mA) to 6mA-dsDNA was markedly decreased (Fig. 5F,G). A

similar result was obtained for the recombinant YTHDF3 W438R/W492R mutant (Appendix Fig. S6C). However, the results of the GST pull-down assay revealed that the W438R/W492R mutation of YTHDF3 did not affect the interaction of YTHDF3 with ALKBH1 (Fig. 5H). Therefore, the MT6mA mutant of YTHDF3 bound to ALKBH1 but did not bind to 6mA in DNA. Finally, we analyzed the sublocalization of YTHDF3 in various cells and tissues and found that YTHDF3 was sublocated in the cytoplasm and nucleus (Fig. 5I; Appendix Fig. S6D,E). Taken together, these findings indicate that YTHDF3 recognizes and binds to 6mA in genomic DNA through the W438/W492 residues in its YTH domain.

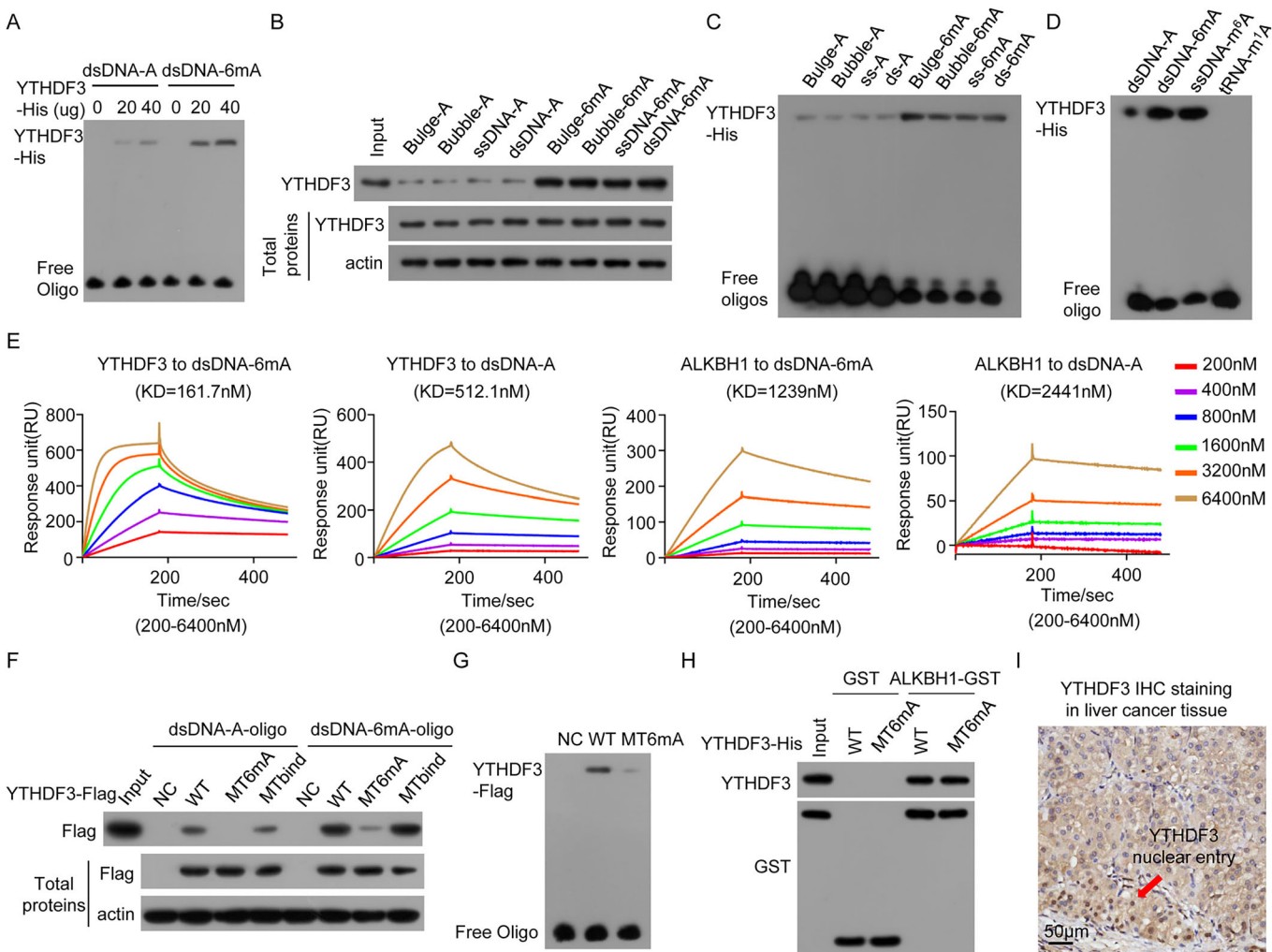

**Figure 5. Compared with ALKBH1, YTHDF3 preferentially directly recognizes and binds to 6mA in genomic DNA.**

(A) The binding of recombinant YTHDF3 at the indicated dose to 6mA-modified and unmodified dsDNA oligos was determined by EMSA. (B, C) The binding of recombinant YTHDF3 to 6mA-modified and unmodified DNA oligos with different conformations was detected by DNA pulldown (B) and EMSA (C). (D) The binding of recombinant YTHDF3 to 6mA-modified and unmodified dsDNA, m⁶A-modified ssRNA and m¹A-modified tRNA oligos was detected via EMSA. (E) The binding of recombinant YTHDF3 and ALKBH1 at different doses to 6mA-modified and unmodified dsDNA oligos was determined by SRP. (F) The indicated YTHDF3 mutants were transfected into cells, and the binding of the cellular YTHDF3 mutants to 6mA-modified and -unmodified dsDNA oligos was detected via DNA pulldown. (G) The binding of wild-type YTHDF3 and the YTHDF3 MT6mA mutant immunopurified from ALKBH1 KO HeLa cells to 6mA-dsDNA oligos was evaluated by EMSA. (H) The direct binding of wild-type YTHDF3 and the YTHDF3 MT6mA mutant to ALKBH1 was analyzed via a GST pull-down assay. (I) The sublocalization of YTHDF3 in liver cancer tissue was analyzed by IHC using an anti-YTHDF3 antibody. Source data are available online for this figure.

## YTHDF3-binding sites highly overlap with ALKBH1-binding sites in the genome

We performed YTHDF3 and ALKBH1 ChIP-seq and compared the overlap of their binding sites in the genome. We found that 60% (79,387 of 132,135 peaks) of the ALKBH1-binding peaks overlapped with the YTHDF3-binding peaks in the genome (Fig. 6A). The chromosome distributions of the ALKBH1 ChIP peaks were highly similar to those of the YTHDF3 ChIP peaks (Fig. 6B,C). Our previous study and other studies revealed that N6AMT1 is a genomic DNA 6mA methyltransferase in eukaryotes (Li et al, 2019b; Xiao et al, 2018). We also found that 53% (14,502 of 27,226 peaks) and 67% (18,261 of 27,226 peaks) of the N6AMT1-binding peaks overlapped with the ALKBH1- and YTHDF3-binding peaks,

respectively (Fig. 6A; Appendix Fig. S7A), but only 11% (14,502 of 132,125 peaks) of the ALKBH1-binding peaks overlapped with N6AMT1-bindg peaks, suggesting that other 6mA methytransferases exist in human cells and ALKBH1 removed genomic 6mA modification mediated by N6AMT1 and other 6mA methytransferases. The chromosome distributions of the N6AMT1 ChIP peaks were also highly similar to those of the YTHDF3 and ALKBH1 ChIP peaks (Appendix Fig. S7B). Taken together, these results indicate that the ALKBH1, YTHDF3 and N6AMT1 binding sites highly overlap in the genome.

We analyzed the ALKBH1, YTHDF3 and N6AMT1 ChIP DNA peaks and found that the [C]AGG[C] motif was the most prevalent motif in the ALKBH1, YTHDF3 and N6AMT1 binding peaks (Fig. 6D), and most of these ALKBH1, YTHDF3 and N6AMT1

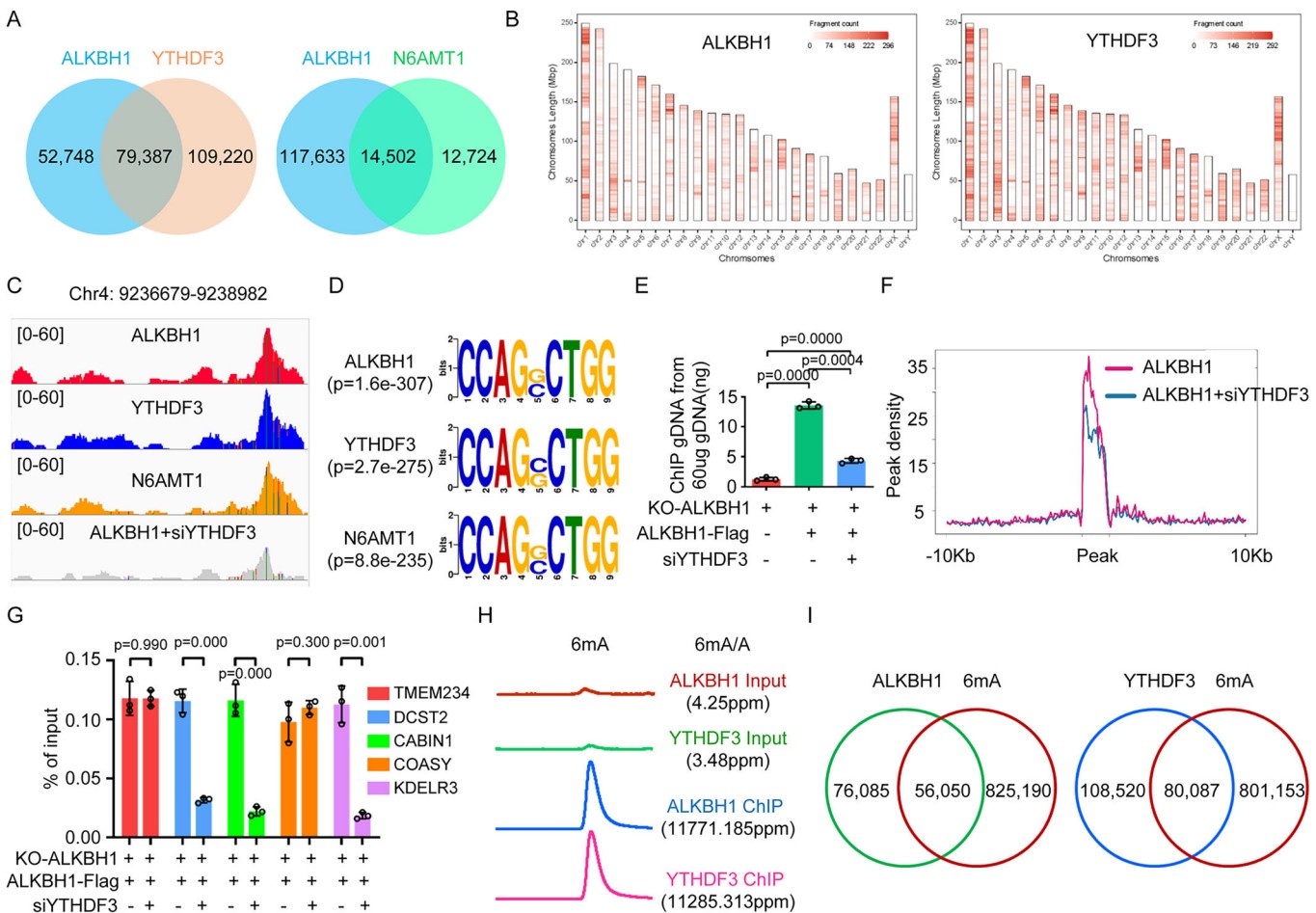

**Figure 6. YTHDF3-binding sites highly overlap with ALKBH1-binding sites in the genome, and YTHDF3- and ALKBH1-binding DNA fragments have greater abundance of 6mA.**

(A) Overlapping between ALKBH1 and YTHDF3 or N6AMT1 ChIP peaks. (B) ALKBH1 and YTHDF3 ChIP peaks across all chromosomes. (C) Representative overlapping regions of the ALKBH1, YTHDF3, N6AMT1 and ALKBH1 with YTHDF3 silencing ChIP peaks. (D) The common motif in the ALKBH1, YTHDF3 and N6AMT1 ChIP peaks. (E) Comparison of the amount of ALKBH1 ChIP DNA between YTHDF3-expressing and YTHDF3-silenced HeLa cells. (F) Cumulative distribution curve for the ALKBH1-binding peak abundance on genomic DNA after YTHDF3 expression was silenced in HeLa cells. (G) Comparison of the relative amount of ChIP DNA from five randomly selected ALKBH1-binding DNA fragments containing 6mA modifications between YTHDF3-expressing and YTHDF3-silenced samples by qPCR. (H) The 6mA/A ratios were measured in the ALKBH1 and YTHDF3 ChIP DNA fragments via LC–MS/MS. (I) Overlapping between ALKBH1 or YTHDF3 ChIP DNA fragments and genomic DNA fragments containing 6mA modifications. Data information: In (E, G), data are presented as mean ± SD. $n = 3$ independent biological replicates, $*P < 0.05$, $**P < 0.01$, $***P < 0.001$, ns, non-significant (Student's $t$ test).

binding peaks were located in intronic and intergenic regions of the human genome (Appendix Fig. S7C), highly similar to the characteristics of genomic 6mA in human genome as shown in our previous report and Appendix Fig. S4.

## YTHDF3 affects the binding of ALKBH1 in the genome

The influence of YTHDF3 on the binding of ALKBH1 in the genome was further investigated. The amount of DNA bound by ALKBH1 was markedly decreased when YTHDF3 expression was knocked down (Fig. 6E). Overall, YTHDF3 silencing led to a decrease in the ALKBH1 binding in the genome (Fig. 6C,F). We further randomly selected five ALKBH1-binding DNA fragments containing a 6mA modification site as representative fragments to investigate the effect of YTHDF3 on ALKBH1 binding in the genome. We found that YTHDF3 silencing decreased ALKBH1

binding to 3 of the 5 ALKBH1-binding DNA fragments (Fig. 6G), indicating that YTHDF3 can affect ALKBH1 binding in the genome and that other unknown regulators of ALKBH1 exist in cells that recruit ALKBH1 to sites near 6mA in genomic DNA.

The amount of DNA bound by YTHDF3 was decreased when N6AMT1 expression was knocked down (Appendix Fig. S7D). We randomly selected five YTHDF3-binding DNA fragments containing a 6mA modification site to investigate the effect of N6AMT1 on YTHDF3 binding in the genome. We found that N6AMT1 silencing decreased YTHDF3 binding to 2 of the 5 YTHDF3-binding DNA fragments (Appendix Fig. S7E), indicating that N6AMT1 can affect YTHDF3 binding in the genome and that the 6mA modifications in some 6mA sites in the genome induced by N6AMT1 are not removed by ALKBH1 and YTHDF3 together, suggesting that other unknown genomic 6mA demethylases and other regulators of ALKBH1 is existed in cells.

## YTHDF3- and ALKBH1-binding DNA fragments highly overlap with DNA fragments containing 6mA in the genome

First, the abundance of 6mA (6mA/A) in the YTHDF3 and ALKBH1 ChIP DNA fragments and the genomic DNA fragment (Input) were determined via LC‑MS/MS. We found that the 6mA abundance of YTHDF3 ChIP DNA was 11,285.314 ppm, while the 6mA abundance of genomic DNA (input) was 3.477 ppm, the 6mA abundance of ALKBH1 ChIP DNA was 11,771.185 ppm, and the 6mA abundance of genomic DNA (input) was 4.248 ppm (Fig. 6H). The abundance of 6mA in the YTHDF3 and ALKBH1 ChIP DNA fragments was 3245 and 2771 greater than that in the genomic DNA fragments (input), respectively (Fig. 6H). These results indirectly indicate that YTHDF3 and ALKBH1 preferentially bind to 6mA in the genome. Furthermore, we found that the binding DNA fragments of ALKBH1, YTHDF3 and N6AMT1 highly overlapped with the DNA fragments containing 6mA modifications in the genome (Fig. 6I; Appendix Fig. S7F). Taken together, these findings indicate that YTHDF3 and ALKBH1 preferentially bind to 6mA in the genome.

## YTHDF3 competes with ALKBH1 to bind 6mA-modified DNA substrates

Different amounts of recombinant ALKBH1 and recombinant wild-type YTHDF3 or its mutant MTbind were coincubated with 6mA-modified dsDNA oligos. A DNA pulldown assay revealed that the binding of YTHDF3 to the 6mA-modified dsDNA increased with increasing YTHDF3, whereas the binding of ALKBH1 (including direct binding and indirect binding through YTHDF3) to the 6mA-modified dsDNA did not change because ALKBH1 could bind to YTHDF3. However, when the binding of YTHDF3 to ALKBH1 was disrupted, the binding of YTHDF3 to the 6mA-modified dsDNA increased with increasing YTHDF3, whereas the binding of ALKBH1 to the 6mA-modified dsDNA decreased because YTHDF3 did not bind to ALKBH1 to recruit ALKBH1 (Fig. 7A). Furthermore, EMSA revealed that the binding of the YTHDF3-ALKBH1 complex to the 6mA-modified dsDNA increased with increasing YTHDF3, whereas the binding of ALKBH1 to the 6mA-modified dsDNA decreased. However, when the binding of YTHDF3 to ALKBH1 was disrupted, the binding of YTHDF3 to the 6mA-modified dsDNA increased, and the binding of ALKBH1 to the 6mA-modified dsDNA decreased with increasing YTHDF3 (Fig. 7B). Collectively, these results indicate that YTHDF3 competes with ALKBH1 to bind the substrate and that compared with ALKBH1, YTHDF3 preferentially binds to 6mA in DNA but YTHDF3 can recruit ALKBH1 near 6mA in DNA by binding to ALKBH1.

The DNA 6mA demethylation activity of ALKBH1 increased with increasing YTHDF3 because YTHDF3 interacts with ALKBH1 and recruits ALKBH1 near 6mA in DNA (the direct binding of ALKBH1 to DNA 6mA decreased, but the indirect binding of ALKBH1 to DNA 6mA increased through YTHDF3) (Appendix Fig. S8A). However, the DNA 6mA demethylation activity of ALKBH1 did not change with increasing YTHDF3 when the interaction of YTHDF3 with ALKBH1 was disrupted and YTHDF3 competitively bound to the 6mA-modified DNA substrate of ALKBH1 and blocked ALKBH1 near 6mA in DNA (Appendix Fig. S8A).

## YTHDF3 recognizes 6mA in genomic DNA and binds to ALKBH1 to recruit ALKBH1 to sites near genomic 6mA modifications to facilitate ALKBH1-mediated removal of 6mA in genomic DNA, including dsDNA

Our results in Fig. 5F–H and Appendix Fig. S6C show that the W438R/W492R mutant (MT6mA) of YTHDF3 binds to ALKBH1 but does not bind to 6mA in dsDNA. Furthermore, compared with wild-type recombinant YTHDF3, the recombinant YTHDF3 MT6mA mutant did not increase the binding of recombinant ALKBH1 to 6mA in dsDNA (Fig. 7C). A similar result was obtained for the MT6mA mutant of YTHDF3 immunopurified from ALKBH1 KO cells (Appendix Fig. S8B). The recombinant YTHDF3 MT6mA mutant did not increase the 6mA demethylase activity of recombinant ALKBH1 in the in vitro demethylation assay (Fig. 7D; Appendix Fig. S8C,D). Similar results were observed for the YTHDF3 MT6mA mutant immunopurified from ALKBH1 KO cells (Appendix Fig. S8E,F). Taken together, these results indicate that YTHDF3 does not increase either the binding of ALKBH1 to 6mA in dsDNA or the 6mA demethylase activity of ALKBH1 when YTHDF3 interacts with ALKBH1 but does not recognize and bind to 6mA in dsDNA.

Our results in Figs. 4H–K and 5F show that the YTHDF3 MTbind mutant binds to 6mA in dsDNA but does not bind to ALKBH1. Furthermore, relative to wild-type YTHDF3, the recombinant YTHDF3 MTbind mutant did not increase the binding of recombinant ALKBH1 to 6mA in dsDNA, although it retained the ability to bind 6mA in dsDNA (Fig. 7E). A similar result was observed for the YTHDF3 MTbind mutant immunopurified from ALKBH1 KO cells (Appendix Fig. S8G). In addition, relative to wild-type YTHDF3, the recombinant YTHDF3 MTbind mutant did not increase the demethylase activity of ALKBH1 toward 6mA in dsDNA (Fig. 7F; Appendix Fig. S8H,I). Similar results were observed for the YTHDF3 MTbind mutant immunopurified from ALKBH1 KO cells (Appendix Fig. S8J,K). Taken together, these results indicate that YTHDF3 does not increase either the binding of ALKBH1 to 6mA in dsDNA or the 6mA demethylase activity of ALKBH1 when YTHDF3 recognizes and binds to 6mA in dsDNA but not to ALKBH1. In summary, YTHDF3 recognizes and binds to 6mA in genomic DNA and binds and recruits ALKBH1 to sites near 6mA modifications to facilitate the removal of 6mA in genomic DNA by ALKBH1.

## Discussion

Previous studies incorporating in vitro reactions and structural analyses have shown that ALKBH1 is a mammalian genomic DNA 6mA demethylase that preferentially removes 6mA in bubbled/bulged DNA and ssDNA, instead of dsDNA (Tian et al, 2020; Wu et al, 2016; Xiao et al, 2018; Zhang et al, 2020). However, ALKBH1 can significantly decrease the 6mA level in cellular genomic DNA (Wu et al, 2016; Xiao et al, 2018; Xie et al, 2018), whose prevailing conformation in living mammalian cells is dsDNA. Therefore, the demethylase activity of ALKBH1 toward 6mA in genomic DNA, especially dsDNA, remains debated. Our findings solve the controversy about the 6mA demethylase of ALKBH1 toward 6mA in genomic dsDNA. A model of the demethylase activity of ALKBH1 toward 6mA in human genomic DNA, especially dsDNA,

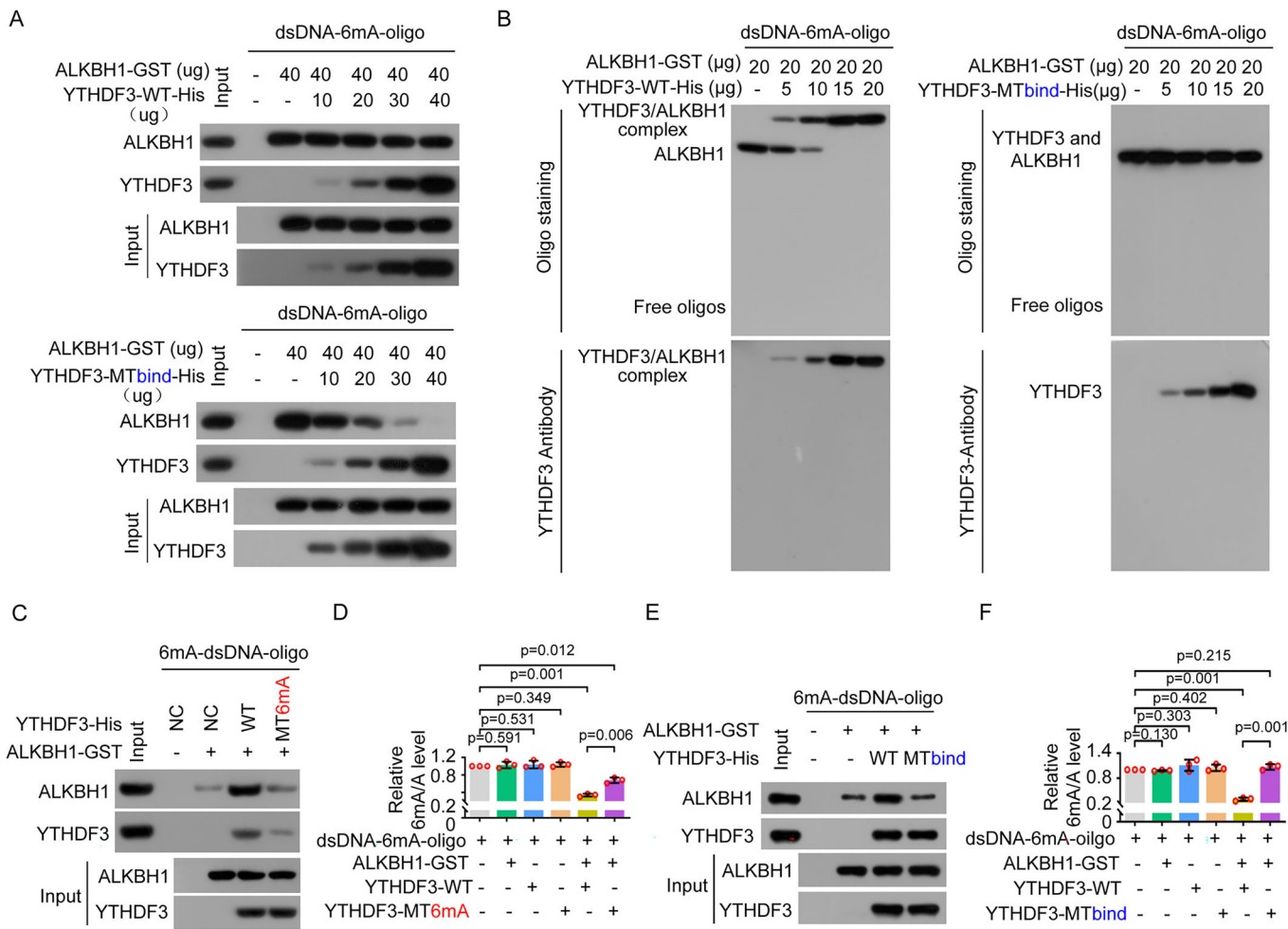

**Figure 7.  YTHDF3 recognizes 6mA in genomic DNA and binds to ALKBH1 to recruit ALKBH1 to sites near 6mA modifications in genomic DNA, thereby facilitating the ALKBH1-mediated removal of 6mA in DNA, including dsDNA.**

(**A, B**) The indicated amounts of recombinant ALKBH1 and recombinant YTHDF3 or its mutant MTbind were incubated with 6mA-dsDNA oligos, and the binding of ALKBH1 and YTHDF3 to 6mA-dsDNA oligos was detected via DNA pull-down (**A**) and EMSA (**B**). (**C**) Recombinant ALKBH1 and/or recombinant wild-type YTHDF3 or the YTHDF3 MT6mA mutant was incubated with 6mA-dsDNA oligos, and the binding of ALKBH1 and YTHDF3 to 6mA-dsDNA was evaluated. (**D**) Recombinant ALKBH1 and/ or recombinant wild-type YTHDF3 or the YTHDF3 MT6mA mutant was incubated with 6mA-dsDNA oligos, and the 6mA levels were determined. (**E**) Recombinant ALKBH1 and/or recombinant wild-type YTHDF3 or the YTHDF3 MTbind mutant was incubated with 6mA-dsDNA oligos, and the binding of ALKBH1 and YTHDF3 to 6mA-dsDNA was evaluated. (**F**) Recombinant ALKBH1 and/or recombinant wild-type YTHDF3 or the YTHDF3 MTbind mutant was incubated with 6mA-dsDNA oligos, and the 6mA levels were determined. Data information: In (**D, F**), data are presented as mean ± SD. $n = 3$ independent biological replicates, *$P < 0.05$, **$P < 0.01$, ***$P < 0.001$, ns, non-significant (Student's $t$ test). Source data are available online for this figure.

was here elucidated. In this model, On the one hand, some ALKBH1 proteins directly preferentially bind to 6mA in bubbled/ bulged DNA and ssDNA relative to dsDNA to remove 6mA in these genomic DNAs. On the other hand, ALKBH1 can catalyze the demethylation of 6mA in various conformations of DNA, including bubbled or bulged DNA, ssDNA and dsDNA, under the guidance of YTHDF3.YTHDF3, as a novel 6mA reader, recognizes and binds to 6mA in genomic DNA, including dsDNA, and binds to ALKBH1 to recruit and guide ALKBH1 to sites near 6mA modifications in genomic DNA, thereby facilitating the ALKBH1-mediated removal of 6mA in genomic DNA, including dsDNA.

YTHDF3 is a member of the YT521-B homology (YTH) domain family of proteins, which comprises YTHDFs (YTHDF1, YTHDF2 and YTHDF3) and YTHDCs (YTHDC1 and YTHDC2), all of

which have been reported to bind directly to m⁶A in RNA and to regulate RNA metabolism as RNA m⁶A readers (Boulias and Greer, 2022; Shi et al, 2017). YTHDF3, as an RNA m⁶A reader, may promote the decay or translation of target mRNAs by directly recognizing m⁶A in RNA through two residues (W438 and W492) in its YTH domain (Chang et al, 2020; Chen et al, 2024). In this study, first, we found that YTHDF3 has a greater ability to bind to 6mA-modified DNA, including bubbled/bulged DNA, ssDNA and dsDNA, than to 6mA-unmodified DNA. Second, compared with ALKBH1, YTHDF3 preferentially recognizes and binds directly to 6mA in DNA. Third, YTHDF3 recognizes 6mA in DNA through the W438/W492 residues in its YTH domain, similar to recognizing m⁶A in RNA, but YTHDF3 does not recognize m¹A in tRNA although ALKBH1 can remove m¹A in tRNA (Liu et al, 2016).

Fourth, YTHDF3-binding sites highly overlap with ALKBH1-binding sites in the genome. Fifth, YTHDF3-binding DNA fragments highly overlap with genomic DNA fragments containing 6mA modifications in the genome. Sixth, the 6mA abundance of YTHDF3-binding DNA fragments is greater than that of genomic DNA. Finally, YTHDF3 is located in the cytoplasm and nucleus and can enter the nucleus in various cells and tissues. Taken together, our results indicate that YTHDF3 is a novel genomic DNA 6mA reader in mammals,

The role of the YTH domain as a 6mA rather than m6A sensor in some cases has precedent in prokaryotes (Woodcock et al, 2020). YTHDF3 can recognize m6A in single-stranded RNA or 6mA in single-stranded DNA with high affinity. However, 6mA is buried in duplex of dsDNA. How YTHDF3 recognizes 6mA in dsDNA will be valuable be further explored in next step by structural analyses.

Our previous study and other studies revealed that N6AMT1 is a genomic DNA 6mA methyltransferase in eukaryotes (Li et al, 2019b; Xiao et al, 2018). However, some structural analyses revealed that the N6AMT1-TRM112 complex does not function as a DNA 6mA methyltransferase (Li et al, 2019a; Metzger et al, 2019; Woodcock et al, 2019). In this study, we reveal that the N6AMT1-binding sites highly overlap with the ALKBH1- and YTHDF3-binding sites and also highly overlap with DNA 6mA modification sites in the genome. And the most prevalent motif [C]AGG[C] in N6AMT1-binding DNA fragments is same with that in ALKBH1- and YTHDF3-binding DNA fragments and also consistent with the most prevalent motif in the 6mA sites in the human genome. N6AMT1 silencing partially decreases the binding of YTHDF3 to genomics DNA. Taken together, these results support that N6AMT1 is a mammalian genomics DNA 6mA methyltransferase.

We reveal that 50–70% of the N6AMT1-binding sites overlap with the YTHDF3/ALKBH1 binding sites, indicating that most of 6mA methylation mediated by N6AMT1 is removed by YTHDF3/ALKBH1. However, only ~11% of ALKBH1-binding sites overlap with N6AMT1-binding sites, indicating that most of YTHDF3/ALKBH1-removed 6mA sites in the genome are from those mediated by other unknown genomic DNA 6mA methyltransferases, instead of N6AMT1.

In summary, we reveal that YTHDF3 guides ALKBH1 to remove 6mA in human genomic DNA, especially including dsDNA. YTHDF3 is a genomic DNA 6mA reader in mammals and recognizes 6mA in genomic DNA and binds ALKBH1 to recruit ALKBH1 to sites near 6mA modifications in DNA, thereby facilitating the ALKBH1-mediated removal of 6mA in DNA, including dsDNA.

# Methods

### Reagents and tools table

| Reagent/resource | Reference or source | Identifier or catalog number |
| --- | --- | --- |
| **Experimental models** | | |
| Human: HTC-116 | ATCC | Cat#CCL-247; RRID: CVCL_0291 |
| Human: HEK293T | ATCC | Cat#CRL-3216; RRID: CVCL_0063 |
| Human: MHCC-97H | Sun Yat-sen University Cancer Center | N/A; RRID:CVCL_4972 |
| Human: HeLa | ATCC | Cat#CCL-2; RRID:CVCL_0030 |
| **Antibodies** | | |
| ALKBH1 | CST | Cat#39013 |
| YTHDF3 (for western blotting) | Proteintech | Cat#25537-1-AP; RRID: AB_2847817 |
| YTHDF3 (for IHC staining) | Invitrogen | Cat#PA5-76378; RRID: AB_2720105 |
| GST | CST | Cat#2625S; RRID: AB_490796 |
| YTHDF1 | Proteintech | Cat#17479-1-AP; RRID: AB_2217473 |
| YTHDF2 | Proteintech | Cat#24744-1-AP; RRID: AB_2687435 |
| YTHDC1 | CST | Cat#54737S; RRID: AB_2799469 |
| Flag | MBL | Cat#M185-3L; RRID: AB_11123930 |
| HA | MBL | Cat#561; RRID: AB_591839 |
| 6mA (for western blotting) | CST | Cat#56593; RRID: AB_2799515 |
| 6mA (for 6mA-IP-seq) | Synaptic Systems | Cat#202-003; RRID: AB_2279214 |
| β-actin | Bioworld | Cat#BS6007M; RRID: AB_2904238 |
| HRP-Goat anti-mouse IgG (H + L) | Proteintech | Cat#SA00001-1; RRID:AB_2722565 |
| HRP-Goat Anti-Rabbit IgG (H + L) | Proteintech | Cat#SA00001-2; RRID:AB_2722564 |
| AF488-labeled Goat Anti-Mouse IgG (H + L) | Beyotime | Cat#A0428; RRID:AB_2893435 |
| **Oligonucleotides and other sequence-based reagents** | | |
| Bulge-6mA | Tian et al, 2020 | Appendix Table S2 |
| Bubble-6mA | Tian et al, 2020 | Appendix Table S2 |
| ds-6mA/6mA-ds-oligo | Tian et al, 2020 | Appendix Table S2 |
| ss-6mA/6mA-ss-oligo | Tian et al, 2020 | Appendix Table S2 |
| M2-A-oligo | This study | Appendix Table S2 |
| M2-6mA-oligo | This study | Appendix Table S2 |
| M3-A-oligo | This study | Appendix Table S2 |
| M3-6mA-oligo | This study | Appendix Table S2 |
| Bulge-M2-6mA-oligo | This study | Appendix Table S2 |
| Bubble-M2-6mA-oligo | This study | Appendix Table S2 |
| ds-M2-6mA-oligo | This study | Appendix Table S2 |
| ss-M2-6mA-oligo | This study | Appendix Table S2 |
| 6mA-ds-oligo (biotin-labeled) | This study | Appendix Table S5 |
| A-ds-oligo (biotin-labeled) | This study | Appendix Table S5 |
| Bulge-6mA (biotin-labeled) | Tian et al, 2020 | Appendix Table S5 |
| Bubble-6mA (biotin-labeled) | Tian et al, 2020 | Appendix Table S5 |
| ds-6mA (biotin-labeled) | Tian et al, 2020 | Appendix Table S5 |
| ss-6mA (biotin-labeled) | Tian et al, 2020 | Appendix Table S5 |
| Bulge-A (biotin-labeled) | This study | Appendix Table S5 |
| Bubble-A (biotin-labeled) | This study | Appendix Table S5 |
| ds-A (biotin-labeled) | This study | Appendix Table S5 |

| Reagent/resource | Reference or source | Identifier or catalog number |
|---|---|---|
| ss-A (biotin-labeled) | This study | Appendix Table S5 |
| ss-m⁶A-oligo (biotin-labeled) | Zhang et al, 2023 | Appendix Table S5 |
| tRNA-m¹A-oligo (biotin-labeled) | Chen et al, 2019 | Appendix Table S5 |
| siALKBH1 | Xiao et al, 2018 | Appendix Table S3 |
| siYTHDF3#1 | This study | Appendix Table S3 |
| siYTHDF3#2 | This study | Appendix Table S3 |
| siMTHFD1#1 | This study | Appendix Table S3 |
| siMTHFD1#2 | This study | Appendix Table S3 |
| siMMS19#1 | This study | Appendix Table S3 |
| siMMS19#2 | This study | Appendix Table S3 |
| NC siRNA | This study | Appendix Table S3 |
| Primers | This study | Appendix Table S4 |
| **Recombinant DNA** | | |
| pcDNA3.1(+) | Invitrogen | Cat#V790-20 |
| pGEX-6P-1 | Youbio | Cat# VT1258 |
| pET26b | Youbio | Cat# VT1205 |
| YTHDF3-Flag | This study | N/A |
| YTHDF3-HA | This study | N/A |
| YTHDF3-Flag-MUT1 | This study | N/A |
| YTHDF3-Flag-MUT2 | This study | N/A |
| YTHDF3-Flag-MUT3 | This study | N/A |
| YTHDF3-Flag-MUT4 | This study | N/A |
| ALKBH1-Flag | Xiao et al, 2018 | N/A |
| ALKBH1-HA | This study | N/A |
| ALKBH1-Flag-MUT1 | This study | N/A |
| ALKBH1-Flag-MUT2 | This study | N/A |
| ALKBH1-Flag-MUT3 | This study | N/A |
| ALKBH1-Flag-MUT4 | This study | N/A |
| ALKBH1-GST | This study | N/A |
| YTHDF3-His | This study | N/A |
| YTHDF3-MTbind-Flag | This study | N/A |
| YTHDF3-MT6mA-Flag | This study | N/A |
| YTHDF3-MTbind-His | This study | N/A |
| YTHDF3-MT6mA-His | This study | N/A |
| ALKBH1 Cas9/sgRNAs plasmids pGE-4 (pU6gRNA1Cas9puroU6gRNA2) | Genepharma | N/A |
| YTHDF3 Cas9/sgRNAs plasmids pGE-4 (pU6gRNA1Cas9puroU6gRNA2) | Genepharma | N/A |
| **Chemicals, enzymes, recombinant proteins and other reagents** | | |
| DreamTaq Green PCR Master Mix (2X) | Thermo | Cat#K1081 |
| Super-Fidelity DNA Polymerase | Vazyme | Cat#C505 |
| Trizol | Tiangen Biotech | Cat#Y1809 |
| 16% Formaldehyde, Methanol-Free | CST | Cat#12606S |
| RNase A | Tiangen Biotech | Cat#Y1831 |
| Proteinase K | Tiangen Biotech | Cat#W9708 |
| Streptavidin−Agarose | Sigma-Aldrich | Cat#S1638 |
| Protein A/G Plus-Agarose | Santa Cruz | Cat#SC-2003 |
| IPTG | Macklin | Cat#367-93-1 |
| Glutathione High Capacity Magnetic Agarose Beads | Sigmaaldrich | Cat#G0924 |
| LightShift™EMSA Optimization | Thermo | Cat#20148X |

| Reagent/resource | Reference or source | Identifier or catalog number |
|---|---|---|
| DpnII | NEB | Cat# R0543S |
| Hybond-N+ membranes | GE Healthcare | Cat# RPN303B |
| PSQ PVDF Membrane | Immobilon | Cat#ISEQ00010 |
| Diethyl pyrocarbonate | Sigma | Cat#V900882-10ML |
| SYBR® Safe DNA gel stain | Invitrogen | Cat#S33102 |
| Transfer RNAs (tRNA) | Sigma | Cat#10109541001 |
| ATP | CST | Cat#9804S |
| HEPES | Genview | Cat#BH160 |
| ascorbic acid | Sigma-Aldrich | Cat#A5960 |
| (NH4)2Fe(SO4)2·6H2O | Sigma-Aldrich | Cat#203505 |
| α-KG | Sigma-Aldrich | Cat#K1128 |
| methylene blue | Sigma-Aldrich | Cat#M4159 |
| Triton™ X-100 | Sigma | Cat#T8787 |
| DAPI Staining Solution | Beyotime | Cat#C1005 |
| Anti-fluorescence quenching solution | Beyotime | Cat# P0183-3 |
| YTHDF3-Flag | This study | N/A |
| ALKBH1-Flag | This study | N/A |
| ALKBH1-GST | This study | N/A |
| YTHDF3-His | This study | N/A |
| YTHDF3-MTbind-Flag | This study | N/A |
| YTHDF3-MT6mA-Flag | This study | N/A |
| YTHDF3-MTbind-His | This study | N/A |
| YTHDF3-MT6mA-His | This study | N/A |
| **Software** | | |
| GraphPad Prism 8.0 | http://www.graphpad.com/scientific-software/prism/ | N/A |
| Protein Pilot Software v4.5 | https://sciex.com/products/software/proteinpilot-software | N/A |
| Mascot v2.3.02 | http://www.matrixscience.com/mascot_support_v2_3.html | N/A |
| SOAPnuke | https://github.com/BGI-flexlab/SOAPnuke | N/A |
| MACS 2.0 | http://liulab.dfci.harvard.edu/MACS/ | N/A |
| Circos | http://mkweb.bcgsc.ca/circos | N/A |
| MEME | http://meme.nbcr.net | N/A |
| Burrows−Wheeler Aligner | http://maq.sourceforge.net | N/A |
| **Other** | | |
| Mut Express II Fast Mutagenesis Kit | Vazyme | Cat#C214 |
| ClonExpress® II One Step Cloning Kit | Vazyme | Cat#C112 |
| PrimeScript™RT reagent Kit with gDNA Eraser | Takara | Cat#RR047A |
| TIANamp Genomic DNA Kit | Tiangen Biotech | Cat#DP304-02 |
| FLAG Immunoprecipitation Kit | Sigmaaldrich | Cat#FLAGIPT1 |
| His-tag Protein Purification Kit | Beyotime | Cat#P2229S |
| QIAquick PCR Purification Kit | Qiagen | Cat#28106 |
| TIANprep Mini Plasmid Kit | Tiangen Biotech | Cat#DP103-03 |
| truChIP Chromatin Shearing Kit | Covaris | Cat#520127 |
| PCR Amplifier | Applied Biosystems | Cat#Veriti96 |
| Gel Imaging System | Bio-Rad | Cat#Gel Doc TM |

| Reagent/resource | Reference or source | Identifier or catalog number |
|---|---|---|
| High-resolution confocal microscope | Nikon | Cat#A1R + N-STORM |
| Focused Ultrasonicator | Covaris | Cat#M200 |
| Real-time fluorescence quantitative PCR system | Applied Biosystems | Cat#StepOnePlus |
| Qubit™4 | Invitrogen | Cat#Q33238 |
| Positive fluorescence microscope | Nikon | Cat#Eclipse Ni-U |
| Molecular interaction analysis system | Biacore | Cat#X100 |

## Cell lines and tissue samples

The cell lines HCT-116 (CCL-247), HeLa (CCL-2) and HEK293T (CRL-3216) were obtained from the American Type Culture Collection; the MHCC-97H cell line was obtained from Kang Lab at Sun Yat-sen University Cancer Center. These cell lines were cultured under standard conditions. The HCT-116 cell line was cultured in RPMI-1640 medium supplemented with 10% fetal bovine serum (FBS), while the HeLa, HEK293T and MHCC-97H cell lines were cultured in DMEM supplemented with 10% FBS. All the cell lines tested negative for mycoplasma contamination.

## Production of recombinant proteins and protein complexes

Recombinant proteins were produced as previously described (Zhang et al, 2023). In brief, the sequences of ALKBH1-GST were inserted into the pGEX vectors, and the sequences of YTHDF3-His and its mutants YTHDF3-His MT6mA and MTbind and ALKBH1-His or ALKBH1 were inserted into the pET 26b vectors. These vectors were transformed or cotransformed into BL21 (DE3) *E. coli* cells, and the expression of ALKBH1-GST, YTHDF3-His, YTHDF3-His MT6mA or YTHDF3-His MTbind, ALKBH1-His or ALKBH1 was induced by 0.5 mMIPTG. The *E. coli* cells were then collected, lysed and ultrasonicated, and the supernatant fractions were then collected. Recombinant ALKBH1-GST protein-conjugated beads were prepared from the supernatant fractions via glutathione Sepharose (Millipore, G0924), and the recombinant ALKBH1-GST protein was eluted with reduced ʟ-glutathione (Sigma, G4251). Recombinant YTHDF3-His protein and its mutants (YTHDF-His MT6mA and MTbind), ALKBH1-His, and the ALKBH1/YTHDF3-His complex were purified from the supernatant fractions via a His-tag protein purification kit (denaturing resistance) (Beyotime, P2229S).

## Production of immunopurified proteins

Immunopurified ALKBH1, YTHDF3 and the YTHDF3 mutants MT6mA and MTbind were produced as previously described with minor modifications (Xiao et al, 2018). In brief, the ALKBH1-Flag plasmids were transfected into YTHDF3-expressing HeLa or YTHDF3 KO HeLa cells, and the YTHDF3-Flag or YTHDF3-Flag MT6mA or MTbind mutant plasmids were transfected into ALKBH1-expressing HeLa or ALKBH1 KO HeLa cells. ALKBH1-Flag, YTHDF3-Flag and YTHDF3-Flag MT6mA and MTbind mutant proteins were purified by using a Flag Immunoprecipitation Kit (FLAGIPT1, Sigma).

## In vitro DNA 6mA demethylase activity assay

The in vitro DNA 6mA demethylase activity assay was performed as previously described with minor modifications (Xiao et al, 2018). In brief, the reactions were performed in a 10 μl volume of demethylation reaction buffer containing 50 pmol of 6mA-DNA oligos with one single 6mA site, 500 ng of recombinant or immunopurified proteins, 50 μM HEPES (pH =7.0), 50 μM KCl, 1 mM $MgCl_2$, 2 mM ascorbic acid, 1 mM α-KG, and 1 mM $(NH_4)_2Fe(SO_4)_2 \cdot 6H_2O$ at 37 °C for 1 h and stopped with 5 mM EDTA by heating at 95 °C for 10 min. Two microliters of each reaction product was then used for subsequent analyses. The 6mA-DNA oligos used here are listed in Appendix Table S1.

## Genomic DNA extraction

Genomic DNA from human cancer cells and tissues was extracted via the TIANamp Genomic DNA Kit (Tiangen Biotech, China). To eliminate possible contamination with m6A-modified RNA, these DNA samples were further treated with RNase A at 37 °C for 2 h. The concentration and quality of the genomic DNA were determined with a NanoDrop 2000 spectrophotometer.

## Dot blotting for DNA 6mA

Dot blotting for DNA 6mA was performed as previously described (Xiao et al, 2018). In brief, RNase A-treated and denatured genomic DNA was spotted on Amersham Hybond-N+ membranes (GE Healthcare, cat# RPN303B) and air dried for 5 min. The membranes were baked at 80 °C for 30 min, subjected to three rounds of crosslinking by exposure to ultraviolet light (254 nm, 400 mJ/cm²) and blocked with blocking buffer (1% bovine serum albumin (BSA) and 5% milk in 0.05% PBST) for 1.5 h at room temperature. The membrane was incubated with a specific anti-6mA antibody overnight at 4 °C. Antibody-bound 6mA on the membrane was detected via an enhanced chemiluminescence SuperSignal West Pico Kit (Thermo, USA). The loaded DNAs were stained with methylene blue (MB).

## Measurement of the 6mA/A ratio via liquid chromatography coupled with tandem mass spectrometry (LC–MS/MS)

The 6mA/A ratios were measured via LC‒MS/MS as previously described (Xiao et al, 2018). In brief, genomic DNA or synthetic DNA oligonucleotides were digested into single nucleosides. Individual nucleosides were analyzed via LC‒MS/MS on an Agilent 6490 Triple Quadrupole mass spectrometer. Nucleosides were quantified via nucleoside-to-base ion mass transitions of 266.1 to 150.1 for DNA 6mA and 252.1 to 136.1 for A. The 6mA and A concentrations were determined on the basis of the standard curves obtained from the corresponding nucleoside standards. The 6mA/A ratio was then calculated on the basis of the calculated concentrations.

## 6mA-RE-qPCR

The restriction enzyme digestion assay together with quantitative PCR (6mA-RE-qPCR) was performed as previously described with

minor modifications (Fu et al, 2015; Xiao et al, 2018). In brief, genomic DNA or 6mA-modified or unmodified dsDNA oligos were treated with DpnII (NEB, cat# R0543S) restriction enzyme or ddH$_2$O at 25 °C overnight. The digested DNA and nondigested DNA were then purified via a universal DNA purification kit (Tiangen, cat# DP214-03), and the purified DNA was subjected to qPCR via TB Green Premix Ex Taq (Tli RNaseH Plus) (TAKARA). Ct values were determined, and ΔCt values were calculated. The primers and DNA oligos used are listed in Appendix Tables S4 and S2, respectively.

## Steady-state kinetics of ALKBH1 demethylation of 6mA in DNA

Determination of the Km and Kcat values of ALKBH1 and ALKBH1 together with YTHDF3 was performed by keeping a constant ALKBH1 enzyme concentration of 0.5 μM, a keeping a constant YTHDF3 concentration of 0.5 μM, and varying the concentration (0, 2, 4, 6, and 8 μM) of the substrates 6mA-dsDNA, 6mA-ssDNA, 6mA-bubble-DNA and 6mA-bulge-DNA oligos containing the restriction enzyme DpnII cutting motif GATC in which A was 6mA-modified. These 6mA-DNA oligos were then treated with DpnII and determined by 6mA-RE-qPCR. All reactions were performed in triplicate and analyzed by GraphPad 8.0 software with the Michaelis–Menten equation.

## Quantification of 6mA in DNA by 6mA-ELISA

DNA 6mA% values were determined via the MethylFlash m⁶A DNA Methylation ELISA Kit (colorimetric) according to the manufacturer's instructions. In brief, DNA was bound to the wells of the plate. After the removal of unbound DNA by washing, a specific anti-6mA antibody was added to the wells. After incubation with the detection antibody included in the kit, the absorbance was measured on a microplate reader at 450 nm. 6mA-unmodified DNA was used as a negative control. The positive control 6mA-DNA was used to produce the standard curve. The 6mA% values were calculated on the basis of the standard curve.

## Coimmunoprecipitation (co-IP)

Co-IP was performed as previously described (Huang et al, 2017). In brief, co-IP was performed using anti-FLAG or anti-HA antibodies and protein A/G agarose beads (Santa Cruz). Proteins in the coimmunoprecipitated complexes were separated by SDS–PAGE, and the gels were subjected to silver staining for mass spectrometry analysis or used for western blot analysis with the indicated antibodies. For mass spectrometry analysis, the differential gel bands and the corresponding negative gel bands were excised and subjected to in-gel digestion with trypsin.

## Mass spectrometry analysis

Protein identification was performed by mass spectrometry as previously described with minor modifications by Bioinnovation Bio., Shenzhen (Huang et al, 2017). In brief, the digested and extracted peptide mixtures were analyzed via a Triple TOF 6600 system (AB SCIEX, USA). The WIFF raw data files were converted into peak list files via Protein Pilot Software v4.5 (AB SCIEX). Protein identification

was performed via the Mascot (v2.3.02) search engine against the UniProt human protein database (released June 2020) with the default parameters. The FDR was set at 1%, and the number of unique peptides was set at ≥2.

The mass spectrometry data used to identify ALKBH1-interacting proteins have been deposited in the ProteomeXchange Consortium via the iProX partner repository (Ma et al, 2019) with the dataset identifier IPX0006743000/PXD044007.

## RNA interference (RNAi)

siRNAs targeting ALKBH1, YTHDF3, MTHFD1 and MMS19 and the NC siRNA (GenePharma) were transfected into cells via RNAiMAX (Invitrogen) for 48 h (unless otherwise stated). The plasmids and siRNAs were cotransfected into cells via Lipofectamine 2000 (Invitrogen) for 48 h (unless otherwise stated). The siRNA sequences are provided in Appendix Table S2.

## Western blotting

Western blotting was performed as previously described with minor modifications (Meng et al, 2020). In brief, total protein from cancer cells or tissues was prepared, separated via 10% SDS–PAGE and then transferred onto a PVDF membrane. The indicated proteins were detected via the corresponding antibodies and the enhanced chemiluminescence SuperSignal West Pico Kit (Thermo, USA) and then exposed to autoradiographic film in a darkroom.

## RNA extraction and RT–PCR

Total RNA was extracted from cells and tissues by using TRIzol total RNA isolation reagent (Invitrogen). The PrimeScript RT Reagent Kit with gDNA Eraser (Takara) was used for reverse transcription of RNA. The indicated mRNA levels were measured via RT–PCR. The RT–PCR primers used in this study are listed in Appendix Table S3.

## Generation of ALKBH1- and YTHDF3-KO cell lines via CRISPR-Cas9

The ALKBH1 and YTHDF3 Cas9/sgRNAs plasmids pGE-4 (pU6-RNA1Cas9puroU6gRNA2) were obtained from GenePharma (Shanghai, China). The two sgRNA targeting sites in ALKBH1 were 5'-TCAGAGCCGGCCCGGGACCG-3' (site 1) and 5'-GGCCCACG-CAGCCCGTGGCA-3' (site 2). The two sgRNA targeting sites in YTHDF3 were ATAGTGTGCCCCCAGTTAGC (site 1) and GTGGACTATAATGCGTATGC (site 2). These ALKBH1 and YTHDF3 CRISPR-Cas9 plasmids were transfected into HeLa cells with Lipo2000 (Invitrogen). Two days after transfection, the cells were seeded into 96-well plates for single-cell culture to obtain single-cell-derived clones. KO of ALKBH1 and TYTHDF3 was confirmed through PCR and western blot analyses.

## 6mA-IP-seq and bioinformatics analyses

6mA-IP-seq was performed as previously described with minor modifications (Xiao et al, 2018). In brief, genomic DNA was treated with RNase A and sonicated to produce fragments of 150–200 bp. The fragmented genomic DNA was incubated with an anti-6mA antibody

(Synaptic Systems) for 2 h at 4 °C. Protein A/G Plus-Agarose (Santa Cruz), prebound to BSA, was then added to the mixture for an additional 2 h of incubation at 4 °C. After extensive washing with PBS, the bound DNA was eluted from the beads, treated with proteinase K and purified with a QIAquick PCR Purification Kit (QIAGEN). The libraries for input and immunoprecipitated DNA were prepared via the NGS Fast DNA Library Prep Set (Illumina) according to the manufacturer's instructions and sequenced on the Illumina NovaSeq 6000 platform by LC-BIO Biotech, Inc. (Hangzhou, China).

Adapter sequences and low-quality reads in the raw data were removed via SOAPnuke (https://github.com/BGI-flexlab/SOAPnuke). The remaining reads were then aligned to the NCBI human reference genome (hg19) via the Burrows–Wheeler Aligner (BWA) with the default parameters. The 6mA-DNA peaks were called with MACS2 with the default parameters. Genome-wide 6mA profiles across all chromosomes were generated with Circos. The MEME program was used to identify motifs in the 6mA-DNA peaks.

The 6mA-IP-seq raw sequence data reported in this paper have been deposited in the Genome Sequence Archive in the National Genomics Data Center (Chen et al, 2021), China National Center for Bioinformation/Beijing Institute of Genomics, Chinese Academy of Sciences (GSA-Human: HRA005118), which are publicly accessible at https://ngdc.cncb.ac.cn/gsa-human.

## 6mA-DNA pull-down assay

Biotin-labeled DNA oligonucleotides containing A or 6mA were synthesized by GeneScript (China). Cellular nuclear protein extracts were prepared via a Nuclear and Cytoplasmic Protein Extraction Kit (Beyotime). Then, 1 nmol of biotin-labeled DNA oligonucleotides containing A or 6mA was bound to 100 μl of streptavidin–agarose beads (Sigma) overnight at 4 °C with rotation. Then, the cellular nuclear protein extracts, immunopurified proteins or recombinant proteins were added to the DNA-immobilized beads, and the mixture was incubated at 30 °C for 30 min. After thorough washing, the beads were eluted by adding 30 μl of protein loading buffer and boiling for 5 min. The eluted mixtures were then analyzed via western blotting with anti-Flag, anti-ALKBH1 or anti-YTHDF3 antibodies. The sequences of the A-DNA oligos and 6mA-DNA oligos are provided in Appendix Table S4.

## Electrophoretic mobility shift assay (EMSA)

EMSA were performed with a LightShiftEMSA Optimization and Control Kit (Thermo Fisher, 20148X) following the manufacturer's instructions and our previous report with minor modifications (Zhu et al, 2020). In brief, biotin-labeled 6mA-modified or 6mA-unmodified DNA oligos, biotin-labeled $m^6A$-modified RNA or biotin-labeled $m^1A$-modified tRNA, recombinant YTHDF3 protein or its mutants MT6mA and MTbind and recombinant ALKBH1 were incubated at 30 °C for 20 min. Then, 2% glutaraldehyde was added to the mixtures and incubated on ice for 15 min. The DNA/RNA−protein mixtures were separated via a 5% nondenaturing gel and transferred to positively charged nylon membranes (GE Healthcare). The membranes were further crosslinked by UV. The biotin-labeled DNA or RNA oligos were detected with a chemiluminescent nucleic acid detection module kit (Thermo Fisher, 89880). The sequences of the 6mA-modified or unmodified DNA oligos, $m^6A$-modified RNAs and $m^1A$-modified tRNAs are listed in Appendix Table S4.

## Surface plasmon resonance (SPR)

SPR was performed using a Biacore X100 plus instrument (Cytiva). Biotin-labeled DNA oligos containing A or 6mA were synthesized by GenScript. The DNA oligos were diluted to 50 μg/ml via Biacore X100 Control Software, injected and immobilized onto the SA sensor chip to achieve an immobilization level of approximately 500 RU (response unit). The first flow cell (control) was prepared without the injection of DNA oligos. The recombinant ALKBH1 or YTHDF3 protein samples were diluted in running buffer (1× HBS-EP+ buffer (Cytiva, BR100669)). Different dilutions of the ALKBH1 or YTHDF3 protein were injected at a flow rate of 30 μl/min for 180 s. Following ALKBH1 or YTHDF3 protein injection, running buffer was passed over the SA sensor surface for 300 s for dissociation. The sensor surface was regenerated by injecting 0.5% SDS solution. The data were analyzed via Biacore X100 evaluation software. These biotin-labeled DNA oligos containing A or 6mA are listed in Appendix Table S4.

## GST pull-down assay

The GST pull-down assay was performed as previously described with minor modifications (Zhang et al, 2023). In brief, recombinant YTHDF3-His proteins were incubated with recombinant ALKBH1-GST protein beads at 4 °C for 1 h. After the protein beads were washed three times, the ALKBH1-GST complexes were eluted from the beads with reduced L-glutathione. YTHDF3-His and ALKBH1-GST were detected via western blotting.

## ChIP and ChIP–qPCR

ChIP was performed using the truChIP Chromatin Shearing Kit (Covaris, 520127) according to the manufacturer's instructions. The cells were crosslinked with 1% formaldehyde (CST, 12606S). Cellular nuclei pellets were prepared, resuspended and sonicated to generate approximately 200 bp chromatin DNA fragments via an ultrasonicator (Covaris, M200). The sheared chromatin was immunoprecipitated with an anti-Flag antibody (CST, 14793S) and Protein A/G Plus-Agarose beads (Santa Cruz) at 4 °C for 2 h. The beads were washed sequentially with low-salt wash buffer (0.1% SDS, 1% Triton X-100, 2 mM EDTA, 20 mM Tris-HCl pH 8.0, and 150 mM NaCl), high-salt wash buffer (0.1% SDS, 1% Triton X-100, 2 mM EDTA, 20 mM Tris-HCl pH 8.0, and 500 mM NaCl), lithium chloride wash buffer (0.25 M LiCl, 1% NP-40, 1% sodium deoxycholate, 1 mM EDTA, and 10 mM Tris-HCl, pH 8.0), and TE buffer (1 mM EDTA and 10 mM Tris-HCl, pH 8.0). The beads were eluted with elution buffer (1% SDS, 100 mM $NaHCO_3$) at 65 °C. The elution mixture was treated with reverse crosslinking buffer from this kit, RNase A, and proteinase K. DNA was purified via the QIAquick® PCR Purification Kit (QIAGEN, 28104). The DNA concentrations were measured via a Qubit™4 instrument (Invitrogen™, Q33238). The obtained DNA was subjected to sequencing or qPCR analysis. The qPCR primers used are listed in Appendix Table S3.

## ChIP-seq and bioinformatics analyses

The ChIP-Seq service was provided by CloudSeq Biotech (Shanghai, China). ChIP-seq libraries were constructed with ChIP and input DNA via the GenSeq® Rapid DNA Library Prep Kit (GenSeq, Inc.) according to the manufacturer's manual. The library quality was determined via an Agilent 2100 Bioanalyzer (Agilent)

and subjected to high-throughput 150-base paired-end sequencing on an Illumina NovaSeq sequencer.

Raw reads were generated after sequencing, image analysis, base calling and quality filtering on a sequencer. Q30 was used to perform quality control. The adaptor sequence was removed via Cutadapt (v1.9.2). These clean reads were aligned to the reference genome (UCSC HG38) via Bowtie2 (v2.2.4) with default parameters. Peak calling was analyzed with MACS (v1.4.2) on the basis of ChIP vs. input. Differentially enriched regions were identified via diffReps (v1.55.6). The enriched peaks were annotated with the latest UCSC RefSeq database to connect the peak information with the gene. The DREME program was used to identify motifs in the ChIP peaks. Peak overlap was performed via the BEDTools (v2.24.0) program. For the overlapping analysis between the ChIP peaks and ChIP peaks, the sequence of the ChIP peaks was extended 500 bp upstream and downstream. The 500 bp DNA fragments upstream and downstream of the 6mA sites from the human genome 6mA data (Xiao et al, 2018) were used to overlap with the ChIP peaks.

The raw ChIP-seq sequence data reported in this paper have been deposited in the Genome Sequence Archive of the National Genomics Data Center (Chen et al, 2021), China National Center for Bioinformation/Beijing Institute of Genomics, Chinese Academy of Sciences (GSA-Human: HRA008265), which are publicly accessible at https://ngdc.cncb.ac.cn/gsa-human.

## IHC assay

IHC staining was performed as previously described using anti-YTHDF3 antibodies (Huang et al, 2017).

## Statistical analysis

Statistical analyses were performed using GraphPad Prism 8 software. Two-tailed unpaired Student's $t$ tests were used for comparisons between two groups. The data are presented as the means $\pm$ SD except where stated otherwise. *$P < 0.05$, **$P < 0.01$ and ***$P < 0.001$ indicate statistically significant values.

# Data availability

The mass spectrometry data have been deposited in the ProteomeXchange Consortium (https://www.proteomexchange.org/) under the accession number IPX0006743000/PXD044007. The 6mA-IP-seq and ChIP-seq data have been deposited in the Genome Sequence Archive in the National Genomics Data Center (https://ngdc.cncb.ac.cn), China National Center for Bioinformation/Beijing Institute of Genomics, Chinese Academy of Sciences under the accession numbers HRA005118 and HRA008265, respectively. All other relevant data are available from the corresponding author G-RY on reasonable request.

The source data of this paper are collected in the following database record: biostudies:S-SCDT-10_1038-S44318-025-00512-2.

# Peer review information

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

## Acknowledgements

This work was supported by National Natural Science Foundation of China (82125029, 32450743 and 82341016), Science and Technology Projects in Guangzhou (2023A03J0390).

## Author contributions

**Xin-Hui Chen**: Data curation; Software; Formal analysis; Investigation; Methodology. **Zi-Lu Wang**: Formal analysis; Investigation. **Jincui Yang**: Investigation. **Min Chen**: Software; Formal analysis; Investigation; Methodology. **Si-Yi Zhao**: Investigation. **Kun-Xiong Guo**: Investigation. **Xuelong Zheng**: Investigation. **Zhengwei Zhao**: Investigation. **Xiaoqiang Chen**: Investigation. **Jing Li**: Investigation. **Min-Min Zhang**: Investigation. **Ling Ran**: Investigation. **Huifang Zhu**: Investigation. **Xiao-Feng Gu**: Investigation;

Methodology. **Guang-Rong Yan**: Conceptualization; Resources; Supervision; Funding acquisition; Methodology; Writing—original draft; Project administration; Writing—review and editing.

Source data underlying figure panels in this paper may have individual authorship assigned. Where available, figure panel/source data authorship is listed in the following database record: biostudies:S-SCDT-10_1038-S44318-025-00512-2.

## Disclosure and competing interests statement
The authors declare no competing interests.

