## [Peer Review File · The EMBO Journal]

YTHDF3 recognizes DNA N6-methyladenine and recruits ALKBH1 for 6mA removal from genomic DNA

Xin-Hui Chen, Zi-Lu Wang, Si-Yi Zhao, Min Chen, Kun-Xiong Guo, Jincui Yang, Xuelong Zheng, Zhengwei Zhao, Xiaoqiang Chen, Jing Li, Min-Min Zhang, Ling Ran, Huifang Zhu, Xiao-Feng Gu, and Guang-Rong Yan

Corresponding author(s): *Guang-Rong Yan (tgryan@jnu.edu.cn)* , *Xiao-Feng Gu (guxiaofeng@caas.cn)*

Review Timeline:

Submission Date:	7th Jan 25
Editorial Decision:	5th Mar 25
Revision Received:	27th May 25
Editorial Decision:	23rd Jun 25
Revision Received:	6th Jul 25
Accepted:	9th Jul 25

Editor: Hartmut Vodermaier

Transaction Report:

Prof. Guang-Rong Yan
Biomedicine Research Center, the Third Affiliated Hospital of Guangzhou Medical University
63 Duobao Road
Guangzhou, Guangdong 510150
China

5th Mar 2025

Re: EMBOJ-2025-120113
YTHDF3 recognizes 6mA in genomic DNA and guides ALKBH1 to remove 6mA in genomic DNA, including dsDNA

Dear Dr. Yan,

Thank you for submitting your study on ALKBH1-YTHDF3 cooperation in 6mA removal to The EMBO Journal. I apologize for the delay in getting back to you with a decision, mainly due to a large submission backlog and very busy reviewers in January. We have now received a complete set of comments from three expert referees, copied below for your information. As you will see, all referees appreciate the interest of the topic and the potential importance of your findings, as well as the unbiased entry point into the study. Nevertheless, they also raise various significant concerns that would need to be clarified before possible publication in a broad general journal like this one.

Should you be able to satisfactorily address the referees' major criticisms, we would be interested in considering a revised manuscript further. This would require better quantification and presentation, as well complementation of biochemical data by some *in vivo* analyses (referee 1); more decisive controls and better biochemical/structural rationalization (referees 2 and 3); and (importantly) strengthening of the mass-spectrometric analyses and ruling out of confounding effects of RNA (referee 3). Since we only allow a single round of major revision, I would encourage you to contact me with a revision plan and preliminary point-by-point response already during the early stages of your revision work, so that we could discuss if and how the main points could best be resolved. I would also be happy to extend the revision time beyond the regular three months if this should be needed. Our 'scooping protection' (meaning that competing work appearing elsewhere in the meantime will not affect our considerations of your study) would of course remain valid also throughout such an extension.

Detailed information on preparing, formatting and uploading a revised manuscript can be found below and in our Guide to Authors. Thank you again for the opportunity to consider this work for The EMBO Journal, and I look forward to hearing from you in due time.

Yours sincerely,

Hartmut Vodermaier

- 3) Revised manuscript text (including main tables, and figure legends for main and EV figures) has to be submitted as editable text file (e.g., .docx format). We encourage highlighting of changes (e.g., via text color) for the referees' reference.
- 4) Each main and each Expanded View (EV) figure should be uploaded as individual production-quality files (preferably in .eps, .tif, .jpg formats). For suggestions on figure preparation/layout, please refer to our Figure Preparation Guidelines: <http://bit.ly/EMBOPressFigurePreparationGuideline>
- 5) Point-by-point response letters should include the original referee comments in full together with your detailed responses to them (and to specific editor requests if applicable), and also be uploaded as editable (e.g., .docx) text files.
- 6) Please complete our Author Checklist, and make sure that information entered into the checklist is also reflected in the manuscript; the checklist will be available to readers as part of the Review Process File. A download link is found at the top of our Guide to Authors: embopress.org/page/journal/14602075/authorguide
- 7) All authors listed as (co-)corresponding need to deposit, in their respective author profiles in our submission system, a unique ORCID identifier linked to their name. Please see our Guide to Authors for detailed instructions.
- 8) Please note that supplementary information at EMBO Press has been superseded by the 'Expanded View' for inclusion of additional figures, tables, movies or datasets; with up to five EV Figures being typeset and directly accessible in the HTML version of the article. For details and guidance, please refer to: embopress.org/page/journal/14602075/authorguide#expandedview
- 9) To facilitate reproducibility and cross-laboratory adoption of methodologies, please structure the Materials & Methods section as outlined in our guide to authors, including a completed Reagents and Tools Table that can be downloaded from our author guidelines as well (<https://www.embopress.org/page/journal/14602075/authorguide#structuredmethods>).
- 10) Digital image enhancement is acceptable practice, as long as it accurately represents the original data and conforms to community standards. If a figure has been subjected to significant electronic manipulation, this must be clearly noted in the figure legend and/or the 'Materials and Methods' section. The editors reserve the right to request original versions of figures and the original images that were used to assemble the figure. Finally, we generally encourage uploading of numerical as well as gel/blot image source data; for details see: embopress.org/page/journal/14602075/authorguide#sourcedata

At EMBO Press, we ask authors to provide source data for the main manuscript figures. Our source data coordinator will contact you to discuss which figure panels we would need source data for and will also provide you with helpful tips on how to upload and organize the files.

In the interest of ensuring the conceptual advance provided by the work, we recommend submitting a revision within 3 months (3rd Jun 2025). Please discuss the revision progress ahead of this time with the editor if you require more time to complete the revisions. Use the link below to submit your revision:

Link Not Available

Referee #1:

In this manuscript, the authors comprehensively investigate the functional link between YTHDF3 and ALKBH1 in regulating the genomic level of 6mA in mammalian cells. They found that of YTHDF3 is important for recruiting ALKBH1 to DNA substrates and genomic targets. This recruitment is dependent on the 6mA binding motif in YTHDF3 in vitro and in vivo. Some of the data is stunning. For example, the remarkable enrichment of the 6mA co-precipitated with YTHDF3 and ALKBH1. Moreover, ALKBH1 can only catalyze 6mA on dsDNA in the presence of YTHDF3. I also appreciate the amount of biochemistry effort although it could be further improved. These data provide a solid framework for the overarching hypothesis. With that said, a number of issues need to be resolved before consideration of a publication in EMBO.

Main comments:

- 1) Although this reviewer appreciates the authors' biochemistry efforts, many of their data is not yet quantified. For example, the kinetics of the ALKBH1 enzyme towards different substrates.
- 2) In regards to N6-AMT-1, as the authors noticed, many structure-function (biochemistry) works don't support its function as a N6-mA DNA methyltransferase. Although it is intriguing it seems to affect ALKBH1 and YTHDF3 function in vivo as presented,

this part of the manuscript is still preliminary. Moreover, including it seems to be a distraction from the main hypothesis rather than supporting it. Hence, I recommend the author to disseminate these results elsewhere.

3) In Figure 7, the dependency of Alkbh1 genomic recruitment on YTHDF3 is only tested by in vitro experiments. This hypothesis can be easily tested by CHIP-seq in vivo.

4) The MS data is quite interesting which serves as a foundation for the whole paper. However, it is not clear from the supplemental table how the significance is calculated and what the control is.

Minor comments:

Although the manuscript is understandable, many grammatical, spelling and style mistakes are present in the current version. For example, on page 15: "... N6AMT1 binds peaks..."

Referee #2:

Overall comments:

=====

The alpha-ketoglutarate dependent dioxygenase ALKBH1 has been described as an 6mA demethylase (for nuclear DNA), however the activity is weak. This suggests that ALKBH1 may require other proteins that guide it to its target site. YTH domains are widely known as m6A sensor domains. However, in prokaryotes, some YTH domains act as 6mA rather than m6A sensors. This finding, together with the weak activity of ALKBH1, makes YTH domain proteins excellent candidates for 6mA sensor proteins that could aid ALKBH1. Here, Chen et al. report that one of the YTH proteins, YTHDF3, indeed has such a role.

It makes the data more convincing that the authors arrived at their model not by a hypothesis driven approach, but were led to the hypothesis by an experimental finding. Contrary to literature reports about only very weak demethylase activity of recombinant ALKBH1, the authors found that their ALKBH1 preparations from human cells were quite active, suggesting either posttranslational modifications not present in the recombinant protein, or ALKBH1 co-purification with interacting partners that enhance the activity. Mass spec analysis identified ~350 candidate interacting proteins, among which the authors selected several proteins. Among these, YTHDF3 turned out to be a protein that:

-directly binds 6mA

-directly binds ALKBH1

-increases the demethylase activity of ALKBH1 towards 6mA at approximately 10% of 6mA sites, half of which are also bound by N6AMT1.

The authors provide very solid evidence for their conclusions, both in cells and in vitro. Given the strength of their evidence, I wondered whether the ALKBH1-YTHDF3 interaction was previously reported in protein interaction databases. However, searching the STRING database with either ALKBH1 or YTHDF3 as the query fails to identify the other protein as an interacting protein. Likewise, the mass spec based hu.MAP 2.0 also fails to indicate the ALKBH1-YTHDF3 interaction, with either protein as the query. Hence, the findings are not only solid and interesting, but also completely novel, not only with respect to the literature, but also with respect to database information.

Comments on merit:

=====

-The authors' title "YTHDF3 recognizes 6mA in genomic DNA and guides ALKBH1 to remove 6mA in genomic DNA, including dsDNA" suggests that the YTHDF3/ALKBH1 pair is a general 6mA demethylase for genomic DNA. However, the author's data in Fig. 6 show that the YTHDF3/ALKBH1 complex has a VERY pronounced sequence specificity for a long target sequence that appears to be also the target (or off-target) of the N6AMT1 methyltransferase. Wouldn't a title that indicates that YTHDF3/ALKBH1 specifically reverses N6AMT1 mediated 6mA methylation be more accurate? Of course, this change of manuscript spin would require an experiment that YTHDF3/ALKBH1 has little effect on nuclear genome DNA methylation in a N6AMT1 KO, but it would greatly strengthen the manuscript.

-Fig. 1E, RE-6mA-qPCR assay: Why are the negative controls in Fig. 1E so different? Shouldn't the negative controls be the same experiment, irrespective of what gene is knocked down in the actual experiment? Have different genes been assessed for different loci? Also, I would have expected that a "Ratio of qPCR product after digestion/no digestion" of 100% corresponds to fully penetrant adenine methylation. The values in Fig. 1E are high already in basal condition, and get close to 1 upon YTHDF3 knockdown. 6mA levels that high seem unlikely. Or is there an exception to the rule that 6mA methylation penetrance is low for sites targeted by N6AMT1? The same question applies also to panels 1F-G, 2B, 2D, 2E, 2F.

-When the YTH domain senses 6mA, it presumably flips the base and scrutinizes it in the pocket for the flipped base. This (or, more generally the footprint of the YTH domain) should physically shield the detected 6mA base from the action of the ALKBH1 dioxygenase. However, the experiment in Fig. 7F suggests otherwise. Do I get it right from the Suppl. that there is only a single

6mA in the experimentally used 6mA-DNA, and that the YTHDF3 enhanced ALKBH1 activity is effective on a single, isolated 6mA (which is the physiologically relevant situation)?

Presentation:

=====

-Introduction: The authors should make it clearer that the role of METTL4 as a DNA methyltransferase is controversial (<https://pubmed.ncbi.nlm.nih.gov/33244833/>).

-Introduction: The authors should make it clear that the role of DMAD/TET as a 6mA demethylase is controversial (<https://pmc.ncbi.nlm.nih.gov/articles/PMC10735219/>)

-Results: Please be clearer what type of DNA is used. is 6mA DNA single stranded or double stranded DNA? How many 6mAs are present?

-Discussion: The role of the YTH domain as a 6mA rather than m6A sensor in some cases has precedent in prokaryotes. The authors could refer to this work to strengthen the plausibility of their conclusions.

Minor:

=====

Legend to Fig. 1: I understand that "MB" in Fig. 1A stands for methylene blue. Please explain this in the legend.

Legend to Fig. 3: Which cell line?

Legend to Fig. 4JK: What does "Input | YTHDF3" stand for?

Results: "60.08% (79387 of 132135 peaks)" should be 60% of peaks. There are so many arbitrary decisions in peak calling that more precision is not reasonable.

Fig. 7: I presume that F3 stands for YTHDF3 and A1 for ALKBH1, however, I see no need to change the nomenclature half-way through the manuscript. If you really insist on doing this, then please explain the abbreviations.

Referee #3:

It is known that ALKBH1 preferentially exhibits 6mA demethylase activity for ssDNA or bubbled/bulged DNA but not for double-stranded DNA (dsDNA). Surprisingly, the authors showed that YTHDF3, an ssRNA m6A reader, increased the 6mA demethylase activity of ALKBH1 in genomic DNA with different conformations, including dsDNA. They showed that YTHDF3 itself prefers to recognizing 6mA in genomic DNA (dsDNA, bubbled/bulged DNA) and speculated that by this mechanism, 6mA-DNA-bound YTHDF3 further binds to ALKBH1 to recruit ALKBH1 to sites near 6mA in genomic DNA, thereby facilitating the ALKBH1-mediated removal of 6mA in genomic DNA. Overall, they provided a line of evidence and concluded that YTHDF3 was a novel genomic dsDNA reader and guided ALKBH1 to remove 6mA in human genomic DNA. It seems interesting. However, due to the debate on the dsDNA 6mA, they should improve their analytical technology to ensure that 6mA contamination is strictly removed.

Major concerns:

1. YTHDF3 can recognize m6A in single-stranded RNA or 6mA in single-stranded DNA with high affinity. Regarding that 6mA is buried in duplex of dsDNA, it is very surprising that YTHDF3 can recognize 6mA in dsDNA. Is there any structural explanation?

2. For LC-MS/MS analysis, a contamination-free approach is required (1. Liu, X. et al. *Cell Res.* 2021, 31: 94-97; 2. Chen, S. et al, *EMBO J.* 2023, e113684.). The authors did not provide details for the measurements and did not explain how they removed the 6mA contamination.

3. Should they use Stable isotope dilution for calibration for the LC-MS/MS detection of 6mA?

4. Besides methylase-deposited 6mA, there is another type of 6mA, misincorporated 6mA. Several papers have reported the presence of misincorporated 6mA (Liu, X. et al. *Cell Res.* 2021, 31: 94-97) and also reported a strict assay for detection of misincorporated 6mA (Liang, Z. et al., *Adv. Sci.* 2024, 2403376.), they should give a brief description on misincorporated 6mA.

5. The authors mainly used Dot Blot for the detection of 6mA, but they did not evaluate the interference from RNA m6A. Essentially, the antibody used for detection of 6mA can also recognize RNA m6A. Moreover, it is very hard to remove all RNA from extracted DNA. Therefore, they should investigate whether RNA m6A interfere with 6mA detection.

Minor:

1. Figure 1B, Pls don't use relative 6mA/dA value, show the ratio of 6mA to dA

2. Figure 1C, They should also show dA peak as a control, a calibration using stable isotope dilution for 6mA is required.

3. Quality control of dsDNA substrates/probes excluding ssDNA is required.

Major revision is recommended

Figure 2I, 6mA abundance, enrichment fold?

Dear Senior Editor, Dr. Vodermaier and Reviewers,

Thank you very much for commenting our manuscript and giving us the opportunity to revise our manuscript entitled “YTHDF3 recognizes 6mA in genomic DNA and guides ALKBH1 to remove 6mA in genomic DNA, including dsDNA”. Your constructive comments and suggestions are helpful and greatly appreciated. Accordingly, we have made the suggested revisions in the manuscript and provided a point-by-point response to these comments as follows. And the revisions were labeled with red. We hope that you will find this revision satisfactory and our revised manuscript acceptable for publication in EMBO J.

Sincerely,

Guang-Rong Yan, PhD

Professor

The Third Affiliated Hospital

Guangzhou Medical University

Reviewer 1:

Question #1:

Although this reviewer appreciates the authors' biochemistry efforts, many of their data is not yet quantified. For example, the kinetics of the ALKBH1 enzyme towards different substrates.

Response:

The steady-state kinetics of the ALKBH1 enzyme towards 6mA-dsDNA, 6mA-ssDNA, 6mA-bubble-DNA and 6mA-bulge-DNA were determined as previously described with modification (Müller TA, et al. Biochemistry. 2017 Apr 4;56(13):1899-1910; Yu D, et al. Nucleic Acids Res. 2021 Nov 18;49(20):11629-11642; Tian LF, et al. Cell Res. 2020 Mar;30(3):272-275.). As shown in revised Figure 2H and 2I, ALKBH1 removed the 6mA modification in 6mA-ssDNA with $K_{cat}=0.0737 \text{ min}^{-1}$ and $K_m=4.835 \mu\text{M}$, 6mA-bubble-DNA with $K_{cat}=0.0933 \text{ min}^{-1}$ and $K_m=2.378 \mu\text{M}$, and 6mA-bluge-DNA with $K_{cat}=0.0969 \text{ min}^{-1}$ and $K_m=2.148 \mu\text{M}$, but ALKBH1 did not remove the 6mA modification in dsDNA. Furthermore, the steady-state kinetics of the ALKBH1 enzyme together with YTHDF3 towards 6mA-dsDNA, 6mA-ssDNA, 6mA-bubble-DNA and 6mA-bulge-DNA were determined. We found that YTHDF3 increased the 6mA demethylation of ALKBH1 toward 6mA in DNA, especially dsDNA and ssDNA.

Substrate	K_m (μM)	K_{cat} (min^{-1})	K_{cat}/K_m ($\text{min}^{-1} \mu\text{M}^{-1}$)	K_m (μM)	K_{cat} (min^{-1})	K_{cat}/K_m ($\text{min}^{-1} \mu\text{M}^{-1}$)
	ALKBH1			ALKBH1+YTHDF3		
6mA-dsDNA	/	/	/	3.643 ± 0.562	0.0498 ± 0.0035	0.014
6mA-ssDNA	4.835 ± 0.896	0.0737 ± 0.0068	0.015	3.225 ± 0.659	0.0981 ± 0.0087	0.030
6mA-bubble-DNA	2.378 ± 0.366	0.0933 ± 0.0056	0.041	2.177 ± 0.393	0.1037 ± 0.0071	0.048
6mA-bluge-DNA	2.148 ± 0.304	0.0969 ± 0.0052	0.045	2.102 ± 0.386	0.1052 ± 0.0072	0.050

Question #2:

In regards to N6-AMT-1, as the authors noticed, many structure-function (biochemistry) works don't support its function as a N6-mA DNA methyltransferase. Although it is intriguing it seems to affect ALKBH1 and YTHDF3 function in vivo as presented, this part of the manuscript is still preliminary. Moreover, including it seems to be a distraction from the main hypothesis rather than supporting it. Hence, I recommend the author to disseminate these results elsewhere.

Response:

It is a good suggestion. I fully agree with your suggestion. This study focused on the regulation of YTHDF3 on genomic 6mA demethylation of ALKBH1. The data about DNA 6mA methyltransferase N6AMT1 were added and provided in this manuscript based on a reviewer's comment when our manuscript was peer-reviewed in other Journal. I also felt that including N6AMT1 was a distraction from the main hypothesis rather than supporting it. I have deleted these results from this manuscript and disseminated these results for publishing in other paper.

Question #3:

In Figure 7, the dependency of Alkbh1 genomic recruitment on YTHDF3 is only tested by in vitro experiments. This hypothesis can be easily tested by CHIP-seq in vivo.

Response:

It is a good suggestion. As shown in Figure 6E, the amount of DNA bound by ALKBH1 was markedly decreased after YTHDF3 expression was silenced when the ChIP DNA amount from the same amount of total genomics DNA were determined. Furthermore, as shown in Figure 6C, we have used ChIP-seq to investigate the dependency of ALKBH1 genomics recruitment on YTHDF3 in vivo after YTHDF3 expression was silenced in cells. The representative peak image about the in vivo effect of YTHDF3 silencing on ALKBH1 genomics recruitment was provided in Figure 6C. I am sorry that the results about the global effect of YTHDF3 on ALKBH1 genomic recruitment in the genome were forgot to provide. As shown in revised

Figure 6F, YTHDF3 silencing decreased the recruitment of ALKBH1 on genome in vivo. The results have added in revised manuscript. Finally, we randomly selected five ALKBH1-binding DNA fragments containing a 6mA modification site as representative fragments to investigate the in vivo effect of YTHDF3 on ALKBH1 binding in the genome. We found that YTHDF3 silencing decreased ALKBH1 binding to 3 of the 5 ALKBH1-binding DNA fragments (Revised Fig. 6G).

Question #4:

The MS data is quite interesting which serves as a foundation for the whole paper. However, it is not clear from the supplemental table how the significance is calculated and what the control is.

Response:

In Appendix Table S1, the significance was automatically calculated by the Mascot (v2.3.02) search engine with the default parameters, which was widely used in protein mass spectrometry identification. I am sorry that I do not also know how the significance is calculated and what the control is in the Mascot (v2.3.02) program.

As shown in Figure 1D, the ALKBH1-Flag vector and blank vector containing Flag Tag were transfected into cells, Co-IP was performed using anti-Flag antibody. The blank vector containing Flag Tag group was as negative control. The differential gel bands and the corresponding negative control gel bands were excised and subjected to in-gel digestion with trypsin and mass spectrometry identification. The proteins interacted with ALKBH1 were obtained after subtracting the proteins identified in the corresponding negative control gel bands (Flag group) from the proteins identified in the differential gel bands (ALKBH1-Flag group).

Question #5:

Although the manuscript is understandable, many grammatic, spelling and style mistakes are present in the current version. For example, on page 15: "... N6AMT1 bindg peaks..."

Response:

Sorry. Our manuscript has been spell-edited by AJE (American Journal Experts). We have sent our manuscript to AJE to re-English-edit.

Reviewer 2:

Question #1:

The authors' title "YTHDF3 recognizes 6mA in genomic DNA and guides ALKBH1 to remove 6mA in genomic DNA, including dsDNA" suggests that the YTHDF3/ALKBH1 pair is a general 6mA demethylase for genomic DNA. However, the author's data in Fig. 6 show that the YTHDF3/ALKBH1 complex has a VERY pronounced sequence specificity for a long target sequence that appears to be also the target (or off-target) of the N6AMT1 methyltransferase. Wouldn't a title that indicates that YTHDF3/ALKBH1 specifically reverses N6AMT1 mediated 6mA methylation be more accurate? Of course, this change of manuscript spin would require an experiment that YTHDF3/ALKBH1 has little effect on nuclear genome DNA methylation in a N6AMT1 KO, but it would greatly strengthen the manuscript.

Response:

Although 53.27% (14502 of 27226 peaks) and 67.07% (18261 of 27226 peaks) of the N6AMT1-binding peaks overlapped with the ALKBH1- and YTHDF3-binding peaks, respectively (old Fig. 6A and Supplementary Fig. S7A), and 89.0% (117633 of 132135 peaks) of ALKBH1-binding peaks did not overlap with N6AMT1-binding peaks, suggesting that about half of 6mA modification induced by N6AMT1 was removed by ALKBH1 and most (about 90%) of ALKBH1-mediated 6mA modification resulted from the 6mA modification induced by some unknown genome 6mA methyltransferases, instead of N6AMT1. Taken together, these results indicate that YTHDF3/ALKBH1 complex regulates 6mA modification induced by N6AMT1 and other 6mA methyltransferases.

The data about DNA 6mA methyltransferase N6AMT1 were added and provided in this manuscript based on a reviewer' comment when our manuscript was peer-reviewed in other Journal. As comments by reviewer#1, our study focused on the regulation of YTHDF3 on genomic 6mA demethylation of ALKBH1, including

N6AMT1 was a distraction from the main hypothesis rather than supporting it. The reviewer#1 recommend the author to disseminate these results elsewhere. We fully agree with the reviewer#1's suggestion. We have deleted the results about N6AMT1 in the revised manuscript and disseminated these results for publishing in other paper.

Question #2:

Fig. 1E, RE-6mA-qPCR assay: Why are the negative controls in Fig. 1E so different? Shouldn't the negative controls be the same experiment, irrespective of what gene is knocked down in the actual experiment? Have different genes been assessed for different loci? Also, I would have expected that a "Ratio of qPCR product after digestion/no digestion" of 100% corresponds to fully penetrant adenine methylation. The values in Fig. 1E are high already in basal condition, and get close to 1 upon YTHDF3 knockdown. 6mA levels that high seem unlikely. Or is there an exception to the rule that 6mA methylation penetrance is low for sites targeted by N6AMT1? The same question applies also to panels 1F-G, 2B, 2D, 2E, 2F.

Response:

Sorry, the part of the results were provided in old Figure 1E. As shown in old Figure 1F (revised Figure 1E), three loci (M2, M3 and M4) were determined in old Figure 1E. The results of each gene of YTHDF3, MTHFD1 and MMS19 were randomly selected from the results of one loci of M2, M3 and M4 to provide in old Figure 1E because the space in Figure 1 was not enough to exhibit all results from three loci. We have revised the old Figure 1E and provided all results of old Figure 1E and transferred all results of old Figure 1E to revised Appendix Figure S1D.

As shown in Supplementary Table S1 in our previous report (Xiao CL, et al. Mol Cell. 2018 Jul 19;71(2):306-318.), we reveal that 6mA modification levels in the different 6mA sites in human genome were different by SMRT-seq. Among them, 6mA levels in some 6mA sites were very high and 6mA levels in about 3.4% of 6mA sites were 100%. In this study, we selected 4 methylated 6mA sites with GATC motifs (M2, M3, M4, and M5 in Supplementary S1E in our previous report (Xiao CL, et al. Mol Cell. 2018 Jul 19;71(2):306-318.) with high 6mA modification level as representative for

determining the changes in the level of 6mA in cellular genomic DNA caused by ALKBH1 or other factors. As shown in supplementary Figure S1E in our previous report, the 6mA level of the four 6mA sites (M2, M3, M4, and M5) were very high. The results in Figure 1F-G, 2B, 2D, 2E, 2F were consistent with our previous results in Supplementary Figure S1E in our previous report.

Supplementary Figure S1E in our previous report (Xiao CL, et al. Mol Cell. 2018 Jul 19;71(2):306-318.).

Question #3:

When the YTH domain senses 6mA, it presumably flips the base and scrutinizes it in the pocket for the flipped base. This (or, more generally the footprint of the YTH domain) should physically shield the detected 6mA base from the action of the ALKBH1 dioxygenase. However, the experiment in Fig. 7F suggests otherwise. Do I get it right from the Suppl. that there is only a single 6mA in the experimentally used 6mA-DNA, and that the YTHDF3 enhanced ALKBH1 activity is effective on a single, isolated 6mA (which is the physiologically relevant situation)?

Response:

As shown in Appendix Table S2 in this manuscript, there is only one single 6mA in the synthetic 6mA-modified dsDNA, ssDNA, bulged DNA and bubbled DNA oligonucleotides used in this study, as previously described in other published papers (Zhang M, et al. Cell Res. 2020 Mar;30(3):197-210; Tian LF, et al. Cell Res. 2020 Mar;30(3):272-275; Xiao CL, et al. Mol Cell. 2018 Jul 19;71(2):306-318; Wu TP, et al. Nature. 2016 Apr 21;532(7599):329-33.).

Based on our previous reports, there were about 880000 6mA sites with the different

6mA levels in human genome, suggesting that on average, there were about one 6mA site with the different 6mA level for every 340bp in human genome. Our study and Other reports revealed that the 6mA/A abundance was 1-3ppm in human cells including cancer cells such as HeLa cells (Shen C, et al. Trends Genet. 2022 May;38(5):454-467.), there might be about one 6mA modification site for every 80Kb-250Kb in human genome if we hypothesize that 6mA modification level in these 6mA sites was 100% in cells with 1-3ppm 6mA/A abundance. Therefore, taken together, one single 6mA site in these synthetic 6mA-modified DNA used in this study and other published papers and one single 6mA site is the physiologically relevant situation.

Question #4:

Introduction: The authors should make it clearer that the role of METTL4 as a DNA methyltransferase is controversial (<https://pubmed.ncbi.nlm.nih.gov/33244833/>).

Response:

I have added this information in the Introduction Section in revised manuscript.

Question #5:

Introduction: The authors should make it clear that the role of DMAD/TET as a 6mA demethylase is controversial (<https://pmc.ncbi.nlm.nih.gov/articles/PMC10735219/>)

Response:

I have added this information in the Introduction Section in revised manuscript.

Question #6:

Results: Please be clearer what type of DNA is used. is 6mA DNA single stranded or double stranded DNA? How many 6mAs are present?

Response:

In this study, the synthesized double-stranded DNA (dsDNA) oligos were mainly used in vitro assays, the related finding were also validated in the synthesized single-stranded DNA (ssDNA), bubbled DNA and bulged DNA oligos in vitro assays.

One single 6mA site was present in these synthesized dsDNA, ssDNA, bubbled DNA and bulged DNA oligos as previous reports (Zhang M, et al. Cell Res. 2020 Mar;30(3):197-210; Tian LF, et al. Cell Res. 2020 Mar;30(3):272-275; Xiao CL, et al. Mol Cell. 2018 Jul 19;71(2):306-318; Wu TP, et al. Nature. 2016 Apr 21;532(7599):329-33.).

We added these information in the revised manuscript.

Question #7:

Discussion: The role of the YTH domain as a 6mA rather than m6A sensor in some cases has precedent in prokaryotes. The authors could refer to this work to strengthen the plausibility of their conclusions.

Response:

It is a good suggestion. We have cited and added this work in the Discussion Section in revised manuscript.

Question #8:

Legend to Fig. 1: I understand that "MB" in Fig. 1A stands for methylene blue. Please explain this in the legend.

Response:

Sorry. The "MB" were labeled in many figures and supplementary Figures. We have added this information in the Materials and Methods Section.

Question #9:

Legend to Fig. 3: Which cell line?

Response:

HeLa cell line, who was widely used as a cell model, was used in Figure 3 because that HeLa cells had a high transfection efficiency. We have added this information in Figure 3 legend.

Question #10:

Legend to Fig. 4JK: What does "Input | YTHDF3" stand for?

Response:

"Input | YTHDF3" in Fig. 4J and 4K stands for recombinant or immunopurified YTHDF3 used before DNA pulldown assay.

Question #11:

Results: "60.08% (79387 of 132135 peaks)" should be 60% of peaks. There are so many arbitrary decisions in peak calling that more precision is not reasonable.

Response:

We have revised it.

Question #12:

Fig. 7: I presume that F3 stands for YTHDF3 and A1 for ALKBH1, however, I see no need to change the nomenclature half-way through the manuscript. If you really insist on doing this, then please explain the abbreviations.

Response:

We have revised F3 and A1 for YTHDF3 and ALKBH1 in Figure 7, respectively.

Reviewer 3:

Question #1:

YTHDF3 can recognize m6A in single-stranded RNA or 6mA in single-stranded DNA with high affinity. Regarding that 6mA is buried in duplex of dsDNA, it is very surprising that YTHDF3 can recognize 6mA in dsDNA. Is there any structural explanation?

Response:

It is a very good question. As reviewer#2' comment #7, the role of the YTH domain as a 6mA rather than m6A sensor in some cases has precedent in prokaryotes, whose DNA prevailing conformation is dsDNA. Because our group are not familiar with structural biology, it's difficult for us to explain this phenomenon from a structural point of view. How YTHDF3 recognizes 6mA in dsDNA will be valuable be further

explored in next step by structural analyses. We have stated this concern in the Discussion section in the revised manuscript.

Question #2:

For LC-MS/MS analysis, a contamination-free approach is required (1. Liu, X. et al. Cell Res. 2021, 31: 94-97; 2. Chen, S. et al, EMBO J. 2023, e113684.). The authors did not provide details for the measurements and did not explain how they removed the 6mA contamination.

Response:

Before we submitted our manuscript, we have read the two papers mentioned (1. Liu, X. et al. Cell Res. 2021, 31: 94-97; 2. Chen, S. et al, EMBO J. 2023, e113684.) and known the stable isotope tracer labeling method for analyzing the genomic 6mA. In the stable isotope tracer labeling method, cells were cultured and labeled with the [15N5]-6mA or [13CD3]-L- Methionine, and 6mA levels in genomic DNA were determined by LC-MS/MS, suggesting that the method is applied for determining change in the level of the *in vivo* cellular genomics DNA 6mA. The stable isotope tracer labeling method is a good method for determining 6mA level change in cellular genomics DNA and avoids the 6mA contamination.

However, considered that the 6mA abundance in cellular genomic DNA is very low in human cells (approx. 1-3ppm in cancer cells), we here applied 6mA-RE-qPCR to determine the change in the level of cellular genomic DNA 6mA induced by ALKBH1 and/or other factors as previously described (Fu Y, et al. Cell. 2015 May 7;161(4):879-892; Xiao CL, et al. Mol Cell. 2018 Jul 19;71(2):306-318.) because the 6mA-RE-qPCR assay can avoid the problems associated with the low abundance of 6mA in cellular genomic DNA, potential prokaryotic DNA contamination in samples, and potential contamination of RNA m⁶A in DNA. 6mA-RE-qPCR is also a contamination-free approach and widely used in many published papers (Fu Y, et al. Cell. 2015 May 7;161(4):879-892; Xiao CL, et al. Mol Cell. 2018 Jul 19;71(2):306-318; Müller TA, et al. Biochemistry. 2017 Apr 4;56(13):1899-1910; Tian LF, et al. Cell Res. 2020 Mar;30(3):272-275.).

The 6mA levels in the synthesized DNA oligos were analyzed by LC-MS/MS. These synthesized DNA containing 6mA modification were free 6mA contamination. In addition, we also applied 6mA-RE-qPCR to determine the kinetics of the ALKBH1 enzyme and ALKBH1 together with YTHDF3 toward 6mA-dsDNA, 6mA-ssDNA, bubbled 6mA-DNA and bulged 6mA-DNA by synthesizing these 6mA-DNA oligos containing the GATC motifs which was restriction enzyme cutting site of Dpn II as previously described (Müller TA, et al. *Biochemistry*. 2017 Apr 4;56(13):1899-1910; Yu D, et al. *Nucleic Acids Res*. 2021 Nov 18;49(20):11629-11642; Tian LF, et al. *Cell Res*. 2020 Mar;30(3):272-275.).

Therefore, both 6mA-RE-qPCR method for 6mA in the cellular genomics DNA and LC-MS/MS method for 6mA in the synthesized DNA oligo used in this study were 6mA contamination-free approach.

Question #3:

Should they use Stable isotope dilution for calibration for the LC-MS/MS detection of 6mA?

Response:

In the stable isotope tracer labeling method, cells were cultured and labeled with the [15N5]-6mA or [13CD3]-L- Methionine, and 6mA levels in genomic DNA were determined by LC-MS/MS, suggesting that the method is applied for determining the *in vivo* change in the level of the cellular genomics DNA 6mA. In this study, the change in the 6mA level in cellular genomics DNA was determined by 6mA-RE-qPCR, which was widely used in many previous reports (Müller TA, et al. *Biochemistry*. 2017 Apr 4;56(13):1899-1910; Yu D, et al. *Nucleic Acids Res*. 2021 Nov 18;49(20):11629-11642; Tian LF, et al. *Cell Res*. 2020 Mar;30(3):272-275.), instead of by LC-MS/MS.

The change in the 6mA level in the *in vitro* synthesized DNA oligos was determined by LC-MS/MS. Therefore, in this study, the non-stable-isotope dilution for calibration for the LC-MS/MS detection of 6mA is feasible.

Question #4:

Besides methylase-deposited 6mA, there is another type of 6mA, misincorporated 6mA. Several papers have reported the presence of misincorporated 6mA (Liu, X. et al. Cell Res. 2021, 31: 94-97) and also reported a strict assay for detection of misincorporated 6mA (Liang, Z. et al., Adv. Sci. 2024, 2403376.), they should give a brief description on misincorporated 6mA.

Response:

It is an important finding. We have added the description on misincorporated 6mA in Introduction section in the revised manuscript.

Question #5:

The authors mainly used Dot Blot for the detection of 6mA, but they did not evaluate the interference from RNA m6A. Essentially, the antibody used for detection of 6mA can also recognize RNA m6A. Moreover, it is very hard to remove all RNA from extracted DNA. Therefore, they should investigate whether RNA m6A interfere with 6mA detection.

Response:

In this study, in addition to dot blotting analyses, the changes in the 6mA level in the *in vivo* cellular genomics DNA were also determined by 6mA-RE-qPCR, which can avoid the RNA and prokaryotic DNA contamination and interfere. The changes in the 6mA level in the *in vitro* synthesized 6mA-DNA oligos, in which there were not RNA m6A interfere, were also determined by LC-MS/MS.

Question #6:

Figure 1B, Pls don't use relative 6mA/dA value, show the ratio of 6mA to dA

Response:

In revised Figure 1C (old Figure 1B), the changes in the 6mA level in the synthetic 6mA-modified DNA oligos with the different conformation including dsDNA, ssDNA, bulged DNA and bubbled DNA were determined after they were treated with recombinant ALKBH1 and immunopurified ALKBH1. Because that the ratio of 6mA

to dA of the four 6mA-modified DNA oligos were different before they were treated with ALKBH1, in order to facilitate the comparison of the change differences of these four 6mA-modified DNA oligos mediated by recombinant ALKBH1 and immunopurified ALKBH1, the relative 6mA/dA value of the four 6mA-modified DNA oligos were provided in the revised Figure 1C (original Figure 1B) as previously described (Zhang M, et al. Cell Res. 2020 Mar;30(3):197-210; Tian LF, et al. Cell Res. 2020 Mar;30(3):272-275; Luo GZ, et al. Nat Commun. 2016 Apr 15;7:11301.).

Question #7:

Figure 1C, They should also show dA peak as a control, a calibration using stable isotope dilution for 6mA is required.

Response:

The dA peak chromatograms have been provided as a control in revised Figure 1B (old Figure 1C).

In Figure 1B and 1C, the changes in 6mA level in the synthesized 6mA-modified DNA oligos including 6mA-dsDNA, 6mA-ssDNA, bulged 6mA-DNA and bubbled 6mA-DNA, instead of the cellular genomics DNA, were determined by LC-MS/MS. These 6mA-DNA oligos were synthesized in vitro, instead of the in vivo cellular genomics DNA from cells who were needed to be cultured and labeled with the [15N5]-6mA or [13CD3]-L- Methionine. Therefore, the calibration analyses using stable-isotope dilution for 6mA were not needed here and the calibration analyses using non-stable-isotope dilution for 6mA used widely were suitable here.

Question #8:

Quality control of dsDNA substrates/probes excluding ssDNA is required.

Response:

It is a good suggestion. Quality control of dsDNA substrates/probes excluding ssDNA was performed using HPLC assay. As shown as follows, The purity of dsDNA was more than 98%.

Question #9:

Figure 2I, 6mA abundance, enrichment fold?

Response:

In revised Figure 2K (old Figure 2I), a representative overlapping region of 6mA-modified DNA peaks in NC, ALKBH1 KO and YTHDF3 KO cells were provided. 6mA abundance/enrichment fold in NC, ALKBH1 KO and YTHDF3 KO cells were provided in revised Figure 2J (old Figure 2H).

Prof. Guang-Rong Yan
Biomedicine Research Center, the Third Affiliated Hospital of Guangzhou Medical University
The Third Affiliated Hospital, Guangzhou Medical University
63 Duobao Road
Guangzhou, Guangdong 510150
China

23rd Jun 2025

Re: EMBOJ-2025-120113R
YTHDF3 recognizes 6mA in genomic DNA and guides ALKBH1 to remove 6mA in genomic DNA, including dsDNA

Dear Dr. Yan,

Thank you for submitting your revised manuscript to The EMBO Journal. Two of the original referees have now assessed it once again (see comments below), and both of them are overall satisfied with the revisions. The only point that continues to concern referee 2 are that more-difficult-to-explain data related to N6AMT1 have been removed instead of followed up on, and maybe this could be alleviated by adding a short caveat paragraph to the discussion, to avoid over-simplification of the study.

In addition, there are still a few editorial issues to be addressed prior to formal acceptance:

- Please make sure to explicitly mention in the legends for BOTH Figure 5I and Appendix Figure S6E that one panel is re-displayed in both of them for reference.
- Please carefully go through the reference list and make sure that each reference is complete with citation year, volume, and page/locator numbers - the latter appear to be missing in at least 9 instances at present.
- Please double-check that all figure panels are called out at least once - e.g. reference to Fig 5B-C seems to currently be missing.
- As we are switching from a free-text author contribution statement towards a more formal statement based on Contributor Role Taxonomy (CRediT) terms, please remove the present Author Contribution section and instead specify each author's contribution(s) directly in the Author Information page of our submission system during upload of the final manuscript. See <https://casrai.org/credit/> for more information.
- In the Data Availability section, please spell out the links for each of the repositories in which data generated with the study are available, please remove the referee access information and instead make sure that all data become openly available at this point.
- Please provide suggestions for a short 'blurb' text prefacing and summing up the study in two sentences (max. 250 characters), followed by 3-5 one-sentence 'bullet points' with brief factual statements of key results of the paper; they will form the basis of an editor-written 'Synopsis' accompanying the online version of the article. Please also upload a synopsis image, which can be used as a "visual title" for the synopsis section of your paper. The image should be in PNG or JPG format with the modest dimensions of EXACTLY 550 pixels wide and 300-600 pixels high.
- Finally, I would propose to simplify and streamline the title as follows, to make it more appealing and clear also for a broad readership:
YTHDF3 recognizes DNA N6-methyladenine and recruits ALKBH1 for 6mA removal from genomic DNA

I am therefore returning the manuscript to you for a final round of revision, to allow you to make the requested modifications and upload the revised files. Once we will have received them, we should be ready to swiftly proceed with formal acceptance and production of the manuscript.

Yours sincerely,

Hartmut Vodermaier

*** PLEASE NOTE: All revised manuscripts are subject to initial checks for completeness and adherence to our formatting guidelines. Revisions may be returned to the authors and delayed in their editorial re-evaluation if they fail to comply to the following requirements (see also our Guide to Authors for further information):

9) To facilitate reproducibility and cross-laboratory adoption of methodologies, please structure the Materials & Methods section as outlined in our guide to authors, including a completed Reagents and Tools Table that can be downloaded from our author guidelines as well (<https://www.embopress.org/page/journal/14602075/authorguide#structuredmethods>).

10) Digital image enhancement is acceptable practice, as long as it accurately represents the original data and conforms to community standards. If a figure has been subjected to significant electronic manipulation, this must be clearly noted in the figure legend and/or the 'Materials and Methods' section. The editors reserve the right to request original versions of figures and the original images that were used to assemble the figure. Finally, we generally encourage uploading of numerical as well as gel/blot image source data; for details see: embopress.org/page/journal/14602075/authorguide#sourcedata

In the interest of ensuring the conceptual advance provided by the work, we recommend submitting a revision within 3 months (21st Sep 2025). Please discuss the revision progress ahead of this time with the editor if you require more time to complete the revisions. Use the link below to submit your revision:

Link Not Available

Referee #1:

The authors provide additional data and analysis which successfully addressed my previous questions, so I don't have any additional major questions. They further elaborate on the structural aspects of the YTHDF3 as most current studies show it only binds to ss substrates. It is very likely this protein stabilizes the ss configuration of the oligonucleotide substrates which shifts the balance.

Referee #2:

As already discussed already in the previous round of reviewing, the concept that the YTH domain containing protein YTHDF3 directs ALKBH1 to server as a 6mA demethylase is convincing.

I will only comment on the issues from the previous revision:

1) Overlap of N6AMT1 and YTHDF3/ALKBH1 sites:

In the earlier version of the manuscript, the authors also had data that the binding sites of YTHDF3/ALKBH1 overlap very strongly (50-70% overlap) with binding sites for N6AMT1. To me, this suggested that YTHDF3/ALKBH1 may be specifically removing N6AMT1 mediated 6mA methylation. I therefore suggested additional experiments to test the effect of YTHDF3/ALKBH1 in an N6AMT1 KO (excepting that the YTHDF3/ALKBH1 pair would make little difference in the KO).

Instead, the authors have now chosen to remove the data on N6AMT1, calling it a "distraction". To me, this seems a bit like removing "inconvenient" data, which are not fully compatible with the spin of the main story. Normally, I would not consent to such "smoothing" of the main story. However, as the authors rightly point out, removing the data on N6AMT1 is indeed suggested by one of the referees (referee 1). Therefore, I am torn about whether or not this way of dealing with the issue is acceptable.

2) Fig. 1E/F, high methylation levels and negative controls:

The issue of negative controls is resolved. The issue of high methylation levels is explained by the preselection of loci that the authors knew to be highly 6mA methylated, and that the authors therefore expected to show strong effects. These loci are not representative for the genome as a whole, but they are good reporters. With this clarification, the new panels Fig. 1E and Fig. 1F are fine.

3) Single or two methylation sites in test substrates, potential YTHDF3/ALKBH1 competition:

I fully agree with the authors that a single 6mA is the physiological situation, the authors estimates of 6mA density in the genome (assuming 100% penetrance) agree with mine. The authors confirm that their test substrates contain only one 6mA site (sorry for missing this information from Table S2). Thus, the potential problem of YTHDF3/ALKBH1 hindering each other by competing for the same site is experimentally ruled out.

My remaining other requests were largely editorial or concerned presentation of figures, and have been addressed by the others.

Dear Senior Editor, Dr. Vodermaier and Reviewers,

Thank you very much for commenting our manuscript and re-giving us the opportunity to revise our manuscript entitled “YTHDF3 recognizes DNA N6-methyladenine and recruits ALKBH1 for 6mA removal from genomic DNA”. Your constructive comments and suggestions are helpful and greatly appreciated. Accordingly, we have made the suggested revisions in the manuscript and provided a point-by-point response to these comments as follows. And the revisions were labeled with red. We hope that you will find this revision satisfactory and our revised manuscript acceptable for publication in EMBO J.

Sincerely,

Guang-Rong Yan, PhD

Professor

The Third Affiliated Hospital

Guangzhou Medical University

Editor Question #1:

Two of the original referees have now assessed it once again (see comments below), and both of them are overall satisfied with the revisions. The only point that continues to concern referee 2 are that more-difficult-to-explain data related to N6AMT1 have been removed instead of followed up on, and maybe this could be alleviated by adding a short caveat paragraph to the discussion, to avoid over-simplification of the study.

Reviewer 2 Question #1:

In the earlier version of the manuscript, the authors also had data that the binding sites of YTHDF3/ALKBH1 overlap very strongly (50-70% overlap) with binding sites for N6AMT1. To me, this suggested that YTHDF3/ALKBH1 may be specifically removing N6AMT1 mediated 6mA methylation. I therefore suggested additional experiments to test the effect of YTHDF3/ALKBH1 in an N6AMT1 KO (excepting that the YTHDF3/ALKBH1 pair would make little difference in the KO).

Instead, the authors have now chosen to remove the data on N6AMT1, calling it a "distraction". To me, this seems a bit like removing "inconvenient" data, which are not fully compatible with the spin of the main story. Normally, I would not consent to such "smoothing" of the main story. However, as the authors rightly point out, removing the data on N6AMT1 is indeed suggested by one of the referees (referee 1). Therefore, I am torn about whether or not this way of dealing with the issue is acceptable.

Response:

As commented as reviewer #2, 50-70% of the N6AMT1-binding sites overlap with the YTHDF3/ALKBH1 binding sites, indicating that most of 6mA methylation mediated by N6AMT1 is removed by YTHDF3/ALKBH1. However, only approximately 11% of ALKBH1-binding sites overlap with N6AMT1-binding sites, indicating that most of YTHDF3/ALKBH1-removed 6mA sites in the genome are from those mediated by other unknown genomic DNA 6mA methyltransferases, instead of N6AMT1. And the results in Appendix Figure S7E also showed that N6AMT1 silencing decreased YTHDF3 binding to 2 of the 5 YTHDF3-binding DNA

fragments containing 6mA site, suggesting that YTHDF3 binding to 3 of the 5 YTHDF3-binding DNA fragments are not affected by N6AMT1. The results indicate that YTHDF3/ALKBH1 pair still effect the genomic DNA 6mA level in N6AMT1 KO cells because that 89% 6mA modification sites mediated by YTHDF3/ALKBH1 pair were not from those mediated by N6AMT1.

We have added the results about N6AMT1 in the second revised manuscript and discussed the issue in the Discussion section.

Editor Question #2:

Please make sure to explicitly mention in the legends for BOTH Figure 5I and Appendix Figure S6E that one panel is re-displayed in both of them for reference.

Response:

Sorry. This re-displayed panel in Appendix Figure S6E has been deleted.

Editor Question #3:

Please carefully go through the reference list and make sure that each reference is complete with citation year, volume, and page/locator numbers - the latter appear to be missing in at least 9 instances at present.

Response:

Sorry. We added and revised them.

Editor Question #4:

Please double-check that all figure panels are called out at least once - e.g. reference to Fig 5B-C seems to currently be missing.

Response:

We have revised them.

Editor Question #5:

As we are switching from a free-text author contribution statement towards a more formal statement based on Contributor Role Taxonomy (CRediT) terms, please

remove the present Author Contribution section and instead specify each author's contribution(s) directly in the Author Information page of our submission system during upload of the final manuscript. See <https://casrai.org/credit/> for more information.

Response:

We have deleted the author contribution statement in revised manuscript.

Editor Question #6:

In the Data Availability section, please spell out the links for each of the repositories in which data generated with the study are available, please remove the referee access information and instead make sure that all data become openly available at this point.

Response:

We have removed the referee access information. All data were openly available.

Editor Question #7:

Please provide suggestions for a short 'blurb' text prefacing and summing up the study in two sentences (max. 250 characters), followed by 3-5 one-sentence 'bullet points' with brief factual statements of key results of the paper; they will form the basis of an editor-written 'Synopsis' accompanying the online version of the article. Please also upload a synopsis image, which can be used as a "visual title" for the synopsis section of your paper. The image should be in PNG or JPG format with the modest dimensions of EXACTLY 550 pixels wide and 300-600 pixels high.

Response:

We have provided these information.

Editor Question #8:

Finally, I would propose to simplify and streamline the title as follows, to make it more appealing and clear also for a broad readership:

YTHDF3 recognizes DNA N6-methyladenine and recruits ALKBH1 for 6mA removal from genomic DNA

Response:

It is a good suggestion. We have revised the title of our manuscript.

Prof. Guang-Rong Yan
Biomedicine Research Center, the Third Affiliated Hospital of Guangzhou Medical University
The Third Affiliated Hospital, Guangzhou Medical University
63 Duobao Road
Guangzhou, Guangdong 510150
China

9th Jul 2025

Re: EMBOJ-2025-120113R1
YTHDF3 recognizes DNA N6-methyladenine and recruits ALKBH1 for 6mA removal from genomic DNA

Dear Dr. Yan,

Thank you for submitting your final revised manuscript for our consideration. I am pleased to inform you that we have now accepted it for publication in The EMBO Journal.

Yours sincerely,

Hartmut Vodermaier
